# DeepNull models non-linear covariate effects to improve phenotypic prediction and association power

Zachary R. McCaw [1,3], Thomas Colthurst[2,3], Taedong Yun [2], Nicholas A. Furlotte[1], Andrew Carroll [1], Babak Alipanahi [1], Cory Y. McLean [2,4✉] & Farhad Hormozdiari [2,4✉]

Genome-wide association studies (GWASs) examine the association between genotype and phenotype while adjusting for a set of covariates. Although the covariates may have non-linear or interactive effects, due to the challenge of specifying the model, GWAS often neglect such terms. Here we introduce DeepNull, a method that identifies and adjusts for non-linear and interactive covariate effects using a deep neural network. In analyses of simulated and real data, we demonstrate that DeepNull maintains tight control of the type I error while increasing statistical power by up to 20% in the presence of non-linear and interactive effects. Moreover, in the absence of such effects, DeepNull incurs no loss of power. When applied to 10 phenotypes from the UK Biobank ($n = 370K$), DeepNull discovered more hits (+6%) and loci (+7%), on average, than conventional association analyses, many of which are biologically plausible or have previously been reported. Finally, DeepNull improves upon linear modeling for phenotypic prediction (+23% on average).

[1] Google Health, Palo Alto, CA, USA. [2] Google Health, Cambridge, MA, USA. [3]These authors contributed equally: Zachary R. McCaw, Thomas Colthurst. [4]These authors jointly supervised this work: Cory Y. McLean, Farhad Hormozdiari. ✉email: cym@google.com; fhormoz@google.com

Genome-wide association studies (GWASs) aim to detect genetic variants or single-nucleotide polymorphisms (SNPs) that are associated with complex traits and diseases. Over the past decade, GWASs have successfully identified thousands of variants associated with various and diverse phenotypes[1–6]. These associations have expanded our knowledge of biological mechanisms[7] and improved our ability to predict phenotypic risk[8].

In most GWAS, the association strength between genotype and phenotype is assessed while adjusting for a set of covariates, such as age, sex, and principal components (PCs) of the genetic relatedness matrix. Covariates are included in GWAS for two main reasons: to increase precision and to reduce confounding. In the linear model setting, adjustment for a covariate will improve precision if the distribution of the phenotype differs across levels of the covariate. For example, when performing GWAS on height, males and females have different means. Adjusting for sex reduces residual variation, and thereby increases power to detect an association between height and the candidate SNPs. Note, however, that omitting sex from the association test is entirely valid. In contrast, omitting a confounder will result in a biased test of association. By definition, a confounder is a common cause of the exposure (i.e. genotype) and the outcome (i.e. phenotype)[9]. In GWAS, a potential confounder is genetic ancestry: two ancestral groups may differ with respect to minor allele frequency (MAF) at common SNPs and, for unrelated reasons, in their phenotypic means. Failure to adjust for ancestry will lead to spurious associations between the phenotype and the SNPs whose MAFs differ across ancestries, inflating the type I error of the association test. To reduce confounding due to population substructure, or the presence of genetically related subgroups within the cohort, multiple genetic PCs are commonly included as covariates during association testing[10,11].

The simplest form of covariate adjustment is to include a linear term for the covariate in the association model. If the phenotypic mean changes non-linearly with the covariate, the residual variation may be further reduced by including higher order adjustments, such as quadratic or interaction terms, as in the following recent examples[12–14]. Shrine et al.[12] included age$^2$ as a covariate when studying chronic obstructive pulmonary disease; Chen et al.[13] included squared body mass index (BMI$^2$) when studying obstructive sleep apnea; and Kosmicki et al.[14] included an age by sex interaction (age × sex) when studying COVID-19 disease outcomes. Although these recent works have recognized the potential importance of modeling non-linear covariate effects, no systematic approach has been described for detecting the appropriate non-linear functions to adjust for in GWAS. The difficulty stems from the exponential number of possible interactions that can arise from a finite set of covariates (e.g. age × sex, age$^2$ × sex, ⋯), and the infinite number of possible transformations of any given continuous covariate (square, logarithm, exponentiation, etc.). Lastly, the optimal number of covariate interactions is not known a priori and requires evaluating different possibilities (Supplementary Table 1).

In this work, we address the issue of model misspecification in GWAS; specifically, misspecification of the relationship between the phenotype and covariates. DeepNull uses a flexible deep neural network (DNN) to learn this potentially complex and non-linear relationship, then adjusts for the network's expectation of the phenotype (based on covariates only) during association testing. Although simpler models (e.g. a second-order interaction model) may suffice in particular cases, the DNN architecture is sufficiently expressive to capture the broad range of phenotype-covariate relationships that researchers might encounter in practice. Moreover, no loss of power is observed when the relationship between the phenotype and covariates is in fact linear.

Using simulated data, we show that DeepNull markedly improves association power and phenotypic prediction in the presence of non-linear covariate effects, and retains equivalent performance in the absence of non-linear effects. We then demonstrate improvements in association power and phenotype prediction across 10 phenotypes from the UK Biobank (UKB)[15], indicating DeepNull's potential for broad utility in biobank-scale GWAS. We provide DeepNull as freely available open-source software (Code Availability) for straightforward integration into existing GWAS association platforms.

## Results

**DeepNull overview.** DeepNull trains a DNN to predict a phenotype of interest from covariates not directly derived from genotypic data (hereafter "non-genetic covariates"). Due to its ability to approximate any continuous mapping[16,17], the DNN can capture complex non-linear relationships between the phenotype and covariates. When performing genetic association testing, the DNN's prediction of the phenotype for each individual is included as a single additional covariate within the association model. Adjusting for the DNN's prediction in the association model is equivalent to regressing it out from both phenotype and genotype. By flexibly modeling the association between phenotype and non-genetic covariates, DeepNull reduces the residual variation, and thereby increases the statistical power (Supplementary Fig. 1, Supplementary Note).

Consider a quantitative phenotype ascertained for a sample of $n$ individuals genotyped at $m$ SNPs. Let $Y = (y_i)_{i=1}^n$ denote the $n \times 1$ phenotype vector, where $y_i$ is the phenotypic value of the $i$th individual; let $G = [g_{ij}]$ denote the $n \times m$ sample by SNP genotype matrix, where $g_{ij}$ is the minor allele count for the ith individual at the jth variant. Let $\bar{G} = [\bar{g}_{ij}] \in \mathbb{R}^{n \times m}$ denote the standardized version of $G$, in which columns have been centered and scaled to have mean zero and unit variance. Furthermore, let $h$ be a (possibly non-linear) function that predicts the phenotype from non-genetic covariates; we learn $h$ using a DNN trained with cross-validation on the sample. The DeepNull association model is as follows:

$$Y = \bar{G}_{.j}\beta_j + \tilde{X}\gamma + H(X)\gamma_h + \varepsilon. \tag{1}$$

Here $\beta_j$ is the effect sizes for the $j$th variant on the phenotype; $\tilde{X} = [x_{ik}]$ is the $n \times (p + g)$ covariate matrix that includes $p$ non-genetic covariates (e.g. age and sex) and $g$ adjustments for genetic confounding (e.g. genetic PCs); $\gamma$ is the $(p + g) \times 1$ vector of association coefficients for all covariates. Compared with the standard GWAS association model, the DeepNull association model differs only by the inclusion of a single additional term $H(X)\gamma_h$: $X$ is the $n \times p$ subset of $\tilde{X}$ consisting of non-genetic covariates (see "Methods"); $H : \mathbb{R}^{n \times p} \to \mathbb{R}^n$ is the function that applies $h$ row-wise to $X$; and $\gamma_h$ is the scalar association coefficient for the DNN's prediction of the phenotype based on non-genetic covariates.

**DeepNull and Baseline perform similarly under linear effects.** We simulated phenotypes based on genotypes and covariates from the UK Biobank[15]. Standardized age, sex, and genotyping_array served as true covariates for 10,000 randomly sampled individuals ("Methods"). First, we considered a linear effect for covariates on phenotypes ($f(x) = \gamma x$). We simulated 100 phenotypes for each of six different genetic architectures with varying amounts of phenotypic variance explained by the genetic data ($\sigma_g^2$) and by covariates ($\sigma_x^2$): (i) $\sigma_g^2 = 0.2$ and $\sigma_x^2 = 0.1$; (ii) $\sigma_g^2 = 0.2$ and $\sigma_x^2 = 0.2$; (iii) $\sigma_g^2 = 0.4$ and $\sigma_x^2 = 0.1$; (iv) $\sigma_g^2 = 0.4$ and $\sigma_x^2 = 0.2$; (v) $\sigma_g^2 = 0.4$ and $\sigma_x^2 = 0.4$; and (vi) $\sigma_g^2 = 0.6$ and

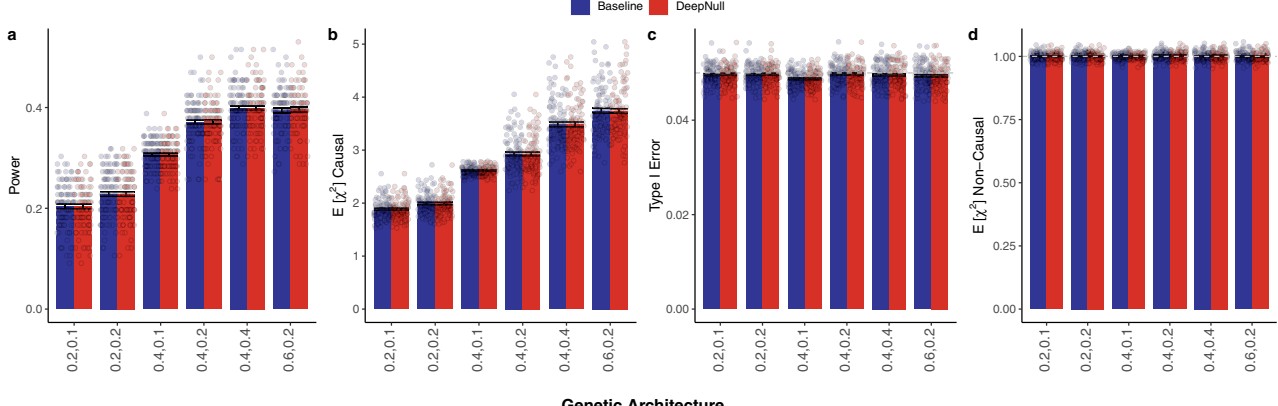

**Fig. 1 DeepNull and baseline model achieve similar results under simulated linear covariate effects. a** Statistical power, and (**b**) expected $\chi^2$ statistics for variants in the causal chromosome (chr22); (**c**) type I error, and (**d**) expected $\chi^2$ statistics for variants on the non-causal chromosomes (chr1 and chr2.). In the case of power and the expected $\chi^2$ statistics in the causal chromosome, higher is better. Methods should have a type I error of 0.05 (gray dashed horizontal line). The expected $\chi^2$ statistics for the non-causal chromosomes should be 1 (gray dashed horizontal line). X-axis values indicate the proportion of phenotypic variance explained by genotypes and covariates, respectively. Error bars are the standard error of the mean for each estimate and and each bar plot summarizes results from $n = 100$ independent simulation replicates. None of the quantities shown is significantly different between Baseline and DeepNull (Wilcoxon signed-rank one-sided test). Source data are provided as a Source Data file.

$\sigma_x^2 = 0.2$. Causal variants were randomly embedded within chr22 and non-causal variants within chr1 and chr2. We compared the DeepNull GWAS with standard GWAS (hereafter referred to as "Baseline"), each of which was performed using BOLT-LMM[18] ("Methods"). Statistical power and expected $\chi^2$ statistics for the causal chromosome (chr22) were similar for DeepNull and Baseline (Fig. 1a, b, Supplementary Table 2). Statistical power for both DeepNull and Baseline increased as genetic heritability $\sigma_g^2$ increased, which is expected since the non-centrality parameter of the $\chi^2$ test increases with the heritability. Additionally, the type I error was maintained at the nominal level, and the expected $\chi^2$ statistics for non-causal variants are similar for both methods (Fig. 1c, d). Thus, DeepNull and Baseline produce similar GWAS results when the effect of the covariates on the phenotype is linear. Lastly, DeepNull and Baseline perform similarly both when excluding non-confounding covariates (i.e., hidden non-confounding covariates, Supplementary Table 3) and when including irrelevant covariates (Supplementary Table 4).

**DeepNull increases power when covariates interact**. We simulated phenotypes using a similar process as described above and used standardized age, sex, genotyping_array, age², age × sex, and age × genotyping_array as true covariates. However, both DeepNull and Baseline are only given age, sex, genotyping_array as known covariates. This simulation setting explores the case where the true covariates are known but their possible interactions are not. DeepNull had higher statistical power (2–13% relative improvement) than baseline, and higher expected $\chi^2$ statistics at causal variants (2–20% relative improvement) across all genetic architectures (Fig. 2a, b, Supplementary Table 5). Importantly, both DeepNull and Baseline control the type I error and generate similar expected $\chi^2$ statistics for non-causal variants (Fig. 2c, d).

**DeepNull increases power under non-linear models**. We simulated phenotypes using a similar process as described above and again used age, sex, genotyping_array, age², age × sex, and age × genotyping_array as true covariates. However, here we fix the genetic architecture ($\sigma_g^2 = 0.4$ and $\sigma_x^2 = 0.4$) and consider non-linear effects of the covariates on the phenotype by using different non-linear functions for $f(\cdot)$ in Eq.

(9): $\sin(x)$, $\exp(x)$, $\log(|x|)$, and sigmoid($x$). Again, both DeepNull and Baseline are only given age, sex, and genotyping_array as known covariates. In all cases, DeepNull outperforms Baseline both in terms of statistical power (3%–9% relative improvement) and expected $\chi^2$ statistics (13%–22% relative improvement), while both methods control the type I error (Supplementary Table 6).

DeepNull is computationally efficient (Supplementary Notes) and its power increases as the sample size increases (Supplementary Notes; Supplementary Fig. 2, Supplementary Table 7). Finally, DeepNull's results are not affected by random seed initialization (Supplementary Notes; Supplementary Fig. 3).

**DeepNull detects more hits than Baseline GWAS on real data**. To explore whether applying DeepNull is beneficial in non-simulated data, we performed GWAS for ten phenotypes from the UK Biobank, using both Baseline and DeepNull. These were: alkaline phosphatase (ALP), alanine aminotransferase (ALT), aspartate aminotransferase (AST), apolipoprotein B (ApoB), calcium, glaucoma referral probability (GRP), LDL cholesterol (LDL), phosphate, sex hormone-binding globulin (SHBG), and triglycerides (TG), each of which has evidence of potentially non-linear relationships between covariates and the phenotype (Supplementary Figs. 4–13). All phenotypes except GRP were extracted directly from the UK Biobank. age, sex, and genotyping_array were considered as input covariates for DeepNull's DNN (Supplementary Table 8). We performed GWAS for these phenotypes using age, sex, genotyping_array, and the top 15 genetic PCs as covariates.

GRP differs from the other phenotypes considered in that it was derived from color retinal fundus images, using the model presented in Alipanahi et al.[19]. As in that study, we are interested in biological signals for glaucoma that are not driven by the vertical cup-to-disc ratio (VCDR). Thus, for GRP only, several additional covariates were included in the association model: VCDRvisit, refractive-error, and image-gradability. To train DeepNull's DNN, we used VCDRvisit, age, sex, and genotyping_array to predict GRP. We then performed GWAS for GRP using age, sex, genotyping_array, the top 15 PCs, VCDRvisit, refractive-error, and image-gradability as covariates.

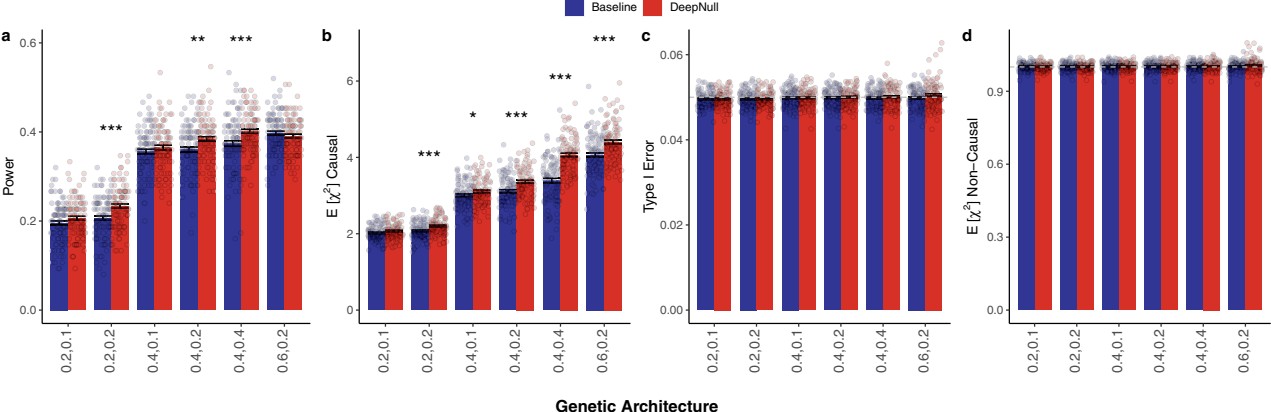

**Fig. 2 DeepNull increases power in the presence of covariate interactions. a** Statistical power, and (**b**) expected $\chi^2$ statistics for variants in the causal chromosome (chr22); (**c**) type I Error, and (**d**) expected $\chi^2$ statistics for variants in the non-causal chromosomes (chr1 and chr2.). In the case of power and expected $\chi^2$ statistics for the causal chromosome, higher is better. Methods should maintain a type I error of no more than 0.05, which is shown by the dashed gray horizontal line. For the non-causal chromosomes, the expected $\chi^2$ statistics should be 1, which is also shown in dashed gray horizontal line. X-axis values indicate the proportion of phenotypic variance explained by genotypes and covariates, respectively. Error bars are the standard error of the mean for each estimate and each bar plot summarizes results from $n = 100$ independent simulation replicates. The numerical results are shown in Supplementary Table 5. Indicators for $P$ value (Wilcoxon signed-rank one-sided test) ranges: [*]$P \leq 0.05$, [**]$P \leq 0.01$, [***]$P \leq 0.001$. Source data are provided as a Source Data file.

**Table 1 DeepNull improves association results relative to the Baseline model on ten phenotypes from the UK Biobank.**

| Pheno | n | #Hits | | %Improve | #Loci | | %Improve |
|---|---|---|---|---|---|---|---|
| | | **Baseline** | **DeepNull** | | **Baseline** | **DeepNull** | |
| ALP | 416,232 | 1697 | **1759** | 3.65% | 336 | **350** | 4.17% |
| ALT | 416,057 | 371 | **379** | 2.16% | 173 | **174** | 0.58% |
| AST | 414,743 | 337 | **351** | 4.15% | 137 | **145** | 5.84% |
| ApoB | 414,639 | 1172 | **1219** | 4.01% | 200 | **217** | 8.50% |
| Calcium | 381,934 | 726 | **739** | 1.79% | 272 | **281** | 3.31% |
| GRP | 65,896 | 28 | **38** | 35.71% | 26 | **38** | 46.15% |
| LDL | 415,892 | 950 | **993** | 4.53% | 193 | **202** | 4.66% |
| Phosphate | 381,362 | 658 | **664** | 0.91% | 224 | **229** | 2.23% |
| SHBG | 378,459 | 1084 | **1120** | 3.32% | 319 | **323** | 1.25% |
| TG | 416,295 | 1221 | **1254** | 2.70% | 261 | **266** | 1.92% |
| Avg. | 370,151 | 824.4 | **851.6** | 6.29% | 214.1 | **222.5** | 7.86% |

*n is the sample size, hits refers to the number of independent genome-wide significant associations detected, and loci is the number of independent regions after merging hits within 250 kb. Bold values in the table indicate the best results.*
*Phenotypic abbreviations: ALP alkaline phosphatase, ALT alanine aminotransferase, AST aspartate aminotransferase, ApoB Apolipoprotein B, GRP glaucoma referral probability, LDL low-density lipoprotein, SHBG sex hormone-binding globulin, TG triglycerides.*

For all GWAS, we first verified that the DeepNull prediction was consistent across all five data folds (Supplementary Table 9). After running GWAS across the entire dataset, we computed the stratified LD score regression (S-LDSC) intercept[20,21] to determine whether there was evidence of inflation due to confounding. In no case did the S-LDSC intercept differ significantly from 1, providing no evidence of inflation due to confounding in our analysis (Supplementary Table 10). In addition, the SNP-heritability of all phenotypes was estimated from both the DeepNull and Baseline summary statistics. For all phenotypes except GRP, the heritability was nominally, though not significantly, greater with DeepNull (Supplementary Table 10).

DeepNull detects more genome-wide significant hits (i.e. independent lead variants) and loci (independent regions after merging hits within 250 kbp together; see Methods) than Baseline for all phenotypes examined (Table 1). For example, we found 46% more significant loci (38 vs. 26) for GRP using DeepNull compared to the Baseline model. Similarly, in the case of LDL, we detected 202 significant loci using DeepNull compared to the 193 significant loci detected with Baseline (4.5% more hits and

4.7% more loci). In addition, 99 of the DeepNull loci were replicated in the GWAS catalog compared with 96 loci for Baseline (Supplementary Fig. 14). For ApoB, DeepNull detected 1219 hits compared to 1172 hits detected by Baseline (4.0% improvement) and DeepNull detected 217 significant loci compared to 200 significant loci obtained from Baseline (8.5% improvement; see Table 1). In addition, 166 of the DeepNull loci were replicated in the GWAS catalog compared with 165 loci for Baseline (Supplementary Fig. 15). For these three phenotypes, we further investigated the biological significance of the detected associations using FUMA[22] (Supplementary Table 11). For GRP, 42 gene sets, predominantly related to pigmentation, were enriched among DeepNull's results, whereas none were enriched among the Baseline results. For LDL, DeepNull detected more gene sets overall (955 Baseline vs. 1000 DeepNull), although the gene sets detected by Baseline scored higher in terms of the average $-\log_{10}(p\text{-value})$ (8.60 Baseline vs. 8.38 DeepNull). However, when focusing on the subset related to lipid metabolism, DeepNull detected more gene sets (65 Baseline vs. 72 DeepNull) and did so at a higher level of significance (average

−log₁₀(p-value): 13.88 Baseline vs. 14.34 DeepNull). For ApoB, DeepNull detected fewer gene sets overall (983 Baseline vs. 946 DeepNull), but at a higher level of significance (average $-\log_{10}$(p-value): 7.65 Baseline vs. 7.81 DeepNull). The gene sets detected by DeepNull related to lipid metabolism and neurological conditions, including Alzheimer's disease.

Overall, the average percentage improvement with DeepNull, taken across phenotypes, was 6.29% for significant hits and 7.86% for loci (Table 1). The average number of hits increased by 3.29%, from 824.4 for Baseline to 851.6 for DeepNull, and the average number of loci increased by 3.93%, from 214.1 to 222.5. In addition, the median number of hits and loci increased by 3.48% and 3.74%, respectively. Lastly, DeepNull tends to have a higher level of significance for variants compared to Baseline (Supplementary Figs. 16–25).

To further understand the source of the DeepNull improvements, we evaluated three additional Baseline models of increasing complexity and a gradient boosted decision tree (GBDT) non-linear model. The first model, which we call "Baseline+ReLU", featurizes `age` into five additional covariates by applying the ReLU function at different thresholds (and solely for GRP, also featurizes `VCDRvisit` in the same way). We observed that while Baseline+ReLU generally identified more significant hits and loci than Baseline, DeepNull consistently outperformed both baseline methods (Supplementary Table 12). The second model, which we call "Second-order Baseline", extends the Baseline model to include all second-order interactions between `age`, `sex`, and `genotyping_array`: $age^2$, `age × sex`, `age × genotyping_array`, and `sex × genotyping_array`. Although the additional second-order interaction covariates consistently improve over the Baseline model results, DeepNull detects as many or more significant loci than Second-order Baseline for nine of the 10 phenotypes (Supplementary Table 13). For AST, LDL, phosphate, and TG, Second-order Baseline and DeepNull detected similar numbers of hits and loci (Supplementary Tables 14 and 15), providing evidence that the hits and loci not found by the Baseline model, which does not include interactions, were in fact true signals. The utility of DeepNull arises because the optimal order of covariate interactions is unknown a priori (Supplementary Table 1), exhaustively enumerating higher order interactions in impractical, and attempting to do so will likely introduce collinearity. Next, we compared the number of hits and loci of DeepNull with an extended Baseline model that performs sex-specific spline fitting (Methods) and observed that DeepNull outperforms this Baseline extension as well (compare Supplementary Tables 14, 16 for hits and Supplementary Tables 15, 17 for loci). Finally, we compared the number of hits and loci of DeepNull with a non-linear GBDT model (Methods) and observed similar numbers of hits and loci (Supplementary Tables 16, 17).

**DeepNull improves phenotype prediction for UKB phenotypes.** An important feature of DeepNull is that it provides additional signal for phenotype prediction. Typically, phenotype prediction models are created using a linear combination of common covariates (such as age and sex) and a polygenic risk score (PRS) defined using GWAS association results. Covariate interactions or higher order terms are occasionally included, but typically in an ad hoc fashion. DeepNull provides a way to easily include potential covariate interactions or higher order terms. The Baseline model includes a PRS computed using PLINK ($PRS_{baseline}$) and linear covariate effects ($PRS_{baseline}$ + Linear covariates). The DeepNull-Baseline model includes a PRS computed in the same way except using association results from DeepNull ($PRS_{DeepNull}$ + Linear covariates), and DeepNull is a model that includes both the DeepNull-based PRS and the DeepNull prediction (non-linear covariate effects).

When compared to the Baseline model, the DeepNull model performs significantly better in terms of the Pearson $R^2$ (Fig. 3). We calculated $R^2$ following previous works[23,24]. We observed that in the case of GRP, LDL, calcium, and ApoB, DeepNull improves phenotype prediction by 83.42%, 40.33%, 23.90%, and 21.61%, respectively. Overall, DeepNull improves phenotype prediction (average improvement = 23.72%, median improvement=16.08%) across the ten phenotypes analyzed (average $n$ = 370K; Supplementary Table 18). In addition, DeepNull has an average $R^2$ of 0.1940 compared to Baseline average $R^2$ of 0.1315 (33.65% improvement; Supplementary Table 18). To determine whether the improved predictive power stems from more accurate GWAS

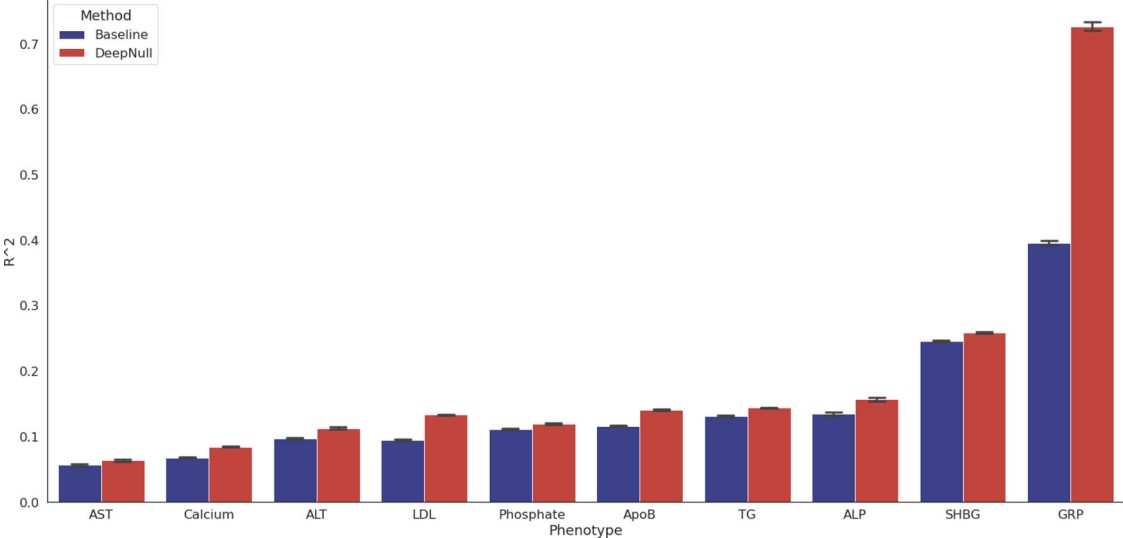

**Fig. 3 DeepNull improves phenotype prediction compared to Baseline.** The X-axis provides the phenotype names and the Y-axis is the $R^2$ where R is Pearson's correlation between true and predicted value of phenotypes. Center of each bar indicates the computed $R^2$ over all samples and the error bars indicate the standard error. Standard errors are computed by performing bootstrapping for each phenotype ($n$ = 1000 bootstrapping trials). Phenotypic abbreviations: alkaline phosphatase (ALP), alanine aminotransferase (ALT), aspartate aminotransferase (AST), apolipoprotein B (ApoB), glaucoma referral probability (GRP), LDL cholesterol (LDL), sex hormone-binding globulin (SHBG), and triglycerides (TG).

effect size estimates or inclusion of the DeepNull DNN prediction, we examined predictive performance of a model that uses `age`, `sex`, and `PRS_DeepNull` ("DeepNull-Baseline"). This model produces slightly higher $R^2$ compared to Baseline for seven of the ten phenotypes, though the difference is not statistically significant for any phenotype (Supplementary Table 18), indicating that most of the improved predictive power arises due to better modeling the effects of non-genetic factors. Next, we compared phenotype prediction of DeepNull to an extended Baseline model that incorporates second-order interactions (additional covariates such as `age²`, `age × sex`, `age × geno-typing_array`). The second-order Baseline model produces similar $R^2$ to DeepNull for many of the phenotypes, but DeepNull increases phenotype prediction of GRP by 11.81% (compare Supplementary Tables 13, 18). Third, we compared phenotype prediction of DeepNull to an extended Baseline model that performs sex-specific spline fitting (Methods) and observed that DeepNull outperforms this Baseline extension as well (compare Supplementary Tables 18, 19). Finally, we compared phenotypic prediction of DeepNull to a non-linear GBDT model ("Methods") and observed similar performance (Supplementary Tables 20, 21).

**DeepNull's covariates should remain in the association model.** When performing genetic association analysis via the model shown in Eq. (1), the covariates $X$ input row-wise to the DNN prediction function $h$ are also included as components of the linear term $\tilde{X}\gamma$. This secondary adjustment for $X$ is necessary because $h$ captures the association between the covariates $X_i$ and the phenotype $y_i$, but does not capture any association between the covariates $X_i$ and genotype $\bar{g}_{ij}$. Failure to include $X_i$ in the final association model is comparable to projecting $X_i$ out of $y_i$ but not $g_{ij}$. To empirically demonstrate the necessity of adjusting $X_i$ in the final association model, we generated phenotypes via

$$y_i = \bar{g}_i\beta + x_i\gamma_1 + x_i^2\gamma_2 + \epsilon_i.$$

For this simulation only, $\bar{g}_i$ was generated as a continuous random variable, allowing for fine control of the correlation between $\bar{g}_i$ and $x_i$, and the model $h$ for predicting $y_i$ from $x_i$ was the oracle model

$$y_i = x_i\gamma_1 + x_i^2\gamma_2 + \epsilon_i.$$

We compare two methods for estimating the genetic effect $\beta$. The unadjusted model incorporates the prediction $h(x_i)$ of $y_i$ based on $x_i$ but omits $x_i$ from the association model, emulating the exclusion of covariates provided to DeepNull from the association model as shown in Eq. (1),

$$y_i = \bar{g}_i\beta + h(x_i)\gamma_h + \epsilon_i. \tag{2}$$

The adjusted model includes both $h(x_i)$ and a linear correction for $x_i$, emulating the application of (1) in practice where the functional form linking $y_i$ and $x_i$ is unknown,

$$y_i = \bar{g}_i\beta + x_i\gamma_1 + h(x_i)\gamma_h + \epsilon_i. \tag{3}$$

Figure 4 presents the relative bias of the unadjusted and linearly adjusted models for estimating the association parameter $\beta$. The relative bias for estimating $\beta$ from the generative model, which represents the best possible performance, is also provided. For these simulations $\gamma_1 = 2$, $\gamma_2 = -1$, and $\beta \in \{\pm 1, \pm 2, \pm 3\}$; the correlation between $\bar{g}_i$ and $x_i$ was 0.5. The unadjusted estimate is generally biased. The magnitude and direction of the bias depend on the coefficients of the generative model. For the unadjusted estimator to be unbiased, $\bar{g}_i$ and $x_i$ must be independent. Since the dependence of $\bar{g}_i$ and $x_i$ is seldom clear, and the linearly adjusted model is unbiased in either case, we adopted the linearly adjusted model for all other analyses. Moreover, the linearly adjusted

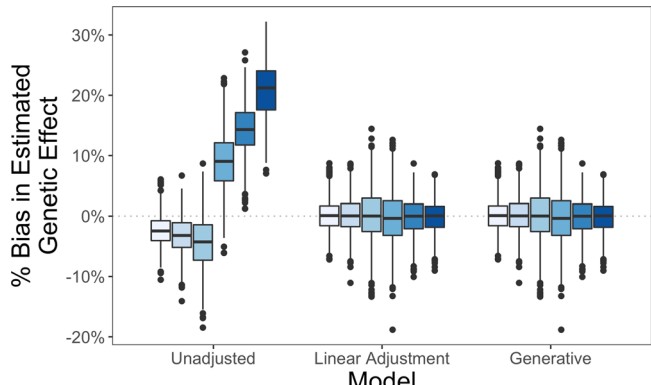

**Fig. 4 Adjusting for covariates provided to DeepNull during association testing is necessary to avoid bias.** The unadjusted model regresses $y_i$ on $\bar{g}_i$ and $h(x_i)$, the prediction of $y_i$ based on $x_i$, omitting $x_i$ from the association model. This approach results in biased estimation of the genetic effect. The linear adjustment model regresses $y_i$ on $\bar{g}_i$, $x_i$, and $h(x_i)$. This approach is unbiased. The generative model regresses $y_i$ on $\bar{g}_i$, $x_i$, and $x_i^2$. This represents the best possible performance. Each box plot summarizes results from $n = 10^3$ independent simulation replicates. The box demarcates, from top to bottom, the 75th, 50th, and 25th percentiles of the corresponding distribution. The whiskers extend between the largest and smallest values within 1.5 times the interquartile range. Any values outside the whiskers are marked by points. Source data are provided as a Source Data file.

estimator remained unbiased in the presence of lower- and higher order covariate effects (Supplementary Figs. 26, 27).

## Discussion
A typical GWAS examines the association between genotypes and the phenotype of interest while adjusting for a set of covariates. While covariates potentially have non-linear effects on the phenotype in many real world settings, due to the challenge of specifying the model, GWAS seldom include non-linear terms. Although it is theoretically possible to model the non-linear effects by considering all possible covariate interactions in a linear model, this approach has multiple limitations. First, the optimal order of covariate interactions is unknown a priori (Supplementary Table 1) as it depends on the particular phenotype and set of covariates. Second, adding higher order covariate interactions requires careful analysis to avoid overfitting and collinearity. We proposed a new framework, DeepNull, that can model the non-linear effect of covariates on phenotypes when such nonlinearity exists. We show that DeepNull can substantially improve phenotype prediction. In addition, we show that DeepNull achieves results similar to a standard GWAS when the effect of covariate on the phenotype is linear, and can significantly outperform a standard GWAS when the covariate effects are nonlinear. DeepNull reduces residual variation, thereby increasing statistical power (Supplementary Fig. 1).

Increasing the statistical power of GWAS is an area of active research that aims to uncover the many variants, each with individually small effect sizes, that collectively explain substantial variation in complex traits and diseases. Multiple complementary approaches have been proposed for increasing statistical power. The most fundamental is to increase the sample size[25]. However, when resources are limited, the sample size cannot be increased indefinitely, and power can be improved through the use of more refined statistical analyses. Linear mixed models (LMMs) were introduced to perform GWASs that include related individuals, who are not statistically independent[18,26–33]. An orthogonal modeling-based approach is to remap or transform the

phenotype to make the distribution of phenotypic residuals more nearly normal[34-38]. While normality of the phenotypic residuals is not necessary for valid association testing, standard association tests are most powerful when the residuals are in fact normally distributed. The final class of methods increases power by leveraging external data on the prior biological plausibility of the variants under study. Highly conserved variants, variants in exons, and protein-coding variants all have higher prior probability of being causal than variants in intergenic regions. A series of methods have been developed that incorporate functional data to detect biologically important variants and up-weight their association statistics or reduce their significance thresholds[39-44]. By focusing on capturing non-linear covariate effects, DeepNull constitutes a distinct approach to improving statistical power of GWAS, one which can be used in combination with any or all of the approaches discussed above.

We note several limitations of our work. First, while training the DeepNull model, we assume individuals (e.g. samples) are independent. Although this is a general assumption among machine learning methods and optimization frameworks, this is not necessarily true in the presence of related individuals. Thus, we believe that an ML optimizer that can incorporate sample relatedness may improve the prediction accuracy of DeepNull's DNN. Importantly, although DeepNull makes the independence assumption during training, this does not mean that type I error is not controlled. Our analyses used BOLT-LMM to perform the association testing, which does correctly account for the relatedness between individuals. Second, DeepNull does not attempt to model possible genotype-covariate ($G \times X$) or genotype-genotype ($G \times G$) interactions. This limitation is shared by standard GWAS and can only be overcome by employing different statistical models that explicitly capture these interactions during association testing. Third, DeepNull's DNN is not easily interpretable compared to less expressive models such as the Baseline model. For improving GWAS power, this is not a major limitation as the parameter of interest is the coefficient describing the relationship between genotype and phenotype. By estimating this coefficient within a linear model that incorporates DeepNull's prediction of phenotype, we obtain a more precise estimate of the genetic effect. For more interpretable phenotypic prediction, possibly at the expense of some prediction accuracy, it may be beneficial to use an alternative non-linear model such as spline regression, generalized additive models[45], symbolic regression[46], or neural additive model[47]. Alternatively, the trained DeepNull model can be interrogated with a variety of methods[48-51], although we note that DNN interpretability is still an active and evolving area of research. Lastly, DeepNull is a proof of concept. For some phenotypes, a simpler model such as the Second-order Baseline model may suffice to capture the phenotype-covariate relationship. For others, an alternative non-linear model such as a GBDT may perform similarly to DeepNull's DNN; for the 10 example UKB phenotypes presented here, a GBDT implemented in XGBoost provided similar performance. Although XGBoost and DNN performed similarly for these phenotypes, the added flexibility of DNNs may prove advantageous for other phenotypes or sets of covariates. For example, DNNs can handle complex inputs such as image and text that XGBoost typically cannot. Importantly, we observed in all cases that DeepNull performed as well or better than current standard practice, and the underlying DNN is sufficiently expressive to capture many of the phenotype-covariate relationships likely to be encountered in practice.

By accurately modeling the non-linear interactions between covariates and the phenotype of interest, DeepNull improved phenotype prediction and association power, both in simulations and on 10 UKB phenotypes. Software for performing end-to-end cross-validated training and prediction is freely available (Code Availability). The resulting phenotypic predictions can readily be included among the input data to commonly-used GWAS models, including PLINK and BOLT-LMM. The improved performance of DeepNull, combined with its ease of use, suggest that it or similar approaches to modeling non-linear covariate effects should become a standard component of performing phenotypic prediction and association testing.

## Methods

*Notation*: We use bold capital letters to indicate matrices, non-bold capital letters to indicate vectors, and non-bold lowercase letters to indicate scalars.

**Standard GWAS**. We consider GWAS of a quantitative trait for a sample of $n$ individuals genotyped at $m$ SNPs. Let $Y = (y_i)_{i=1}^n$ denote the $n \times 1$ phenotype vector, where $y_i$ is the phenotypic value of the $i$th individual, and $G = [g_{ij}]$ the $n \times m$ sample by SNP genotypes matrix, where $g_{ij}$ is the minor allele count for the $i$th individual at the $j$th variant. Since human genomes are diploid, each variant has 3 possible minor allele counts: $g_{ij} \in \{0, 1, 2\}$. $G_{.j} = (g_{ij})_{i=1}^n$ is a vector of minor allele counts for all individuals at the $j$th SNP. For simplicity, assume the phenotypes and genotypes are standardized to have zero mean and unit variance. Let $\bar{G} = [\bar{g}_{ij}] \in \mathbb{R}^{n \times m}$ be the standardized version of $G$, i.e. the empirical mean and variance of $\bar{G}_{.j}$ are zero and one, respectively: $\frac{1}{n} \sum_i \bar{g}_{ij} = 0$ and $\frac{1}{n} \sum_i \bar{g}_{ij}^2 = 1$ for each $j$th SNP.

A typical GWAS assumes the effect of each variant on the phenotype is linear and additive. Thus, we have the following generative model:

$$Y = \bar{G}\beta + X\gamma + \varepsilon \qquad (4)$$

where $\beta$ is the $m \times 1$ vector of effect sizes for each variant on the phenotype, $X = [x_{ik}]$ is the $n \times q$ covariate matrix, including covariates such as age and sex, and $\gamma$ is the $q \times 1$ vector of association coefficients for the covariates. Let $X$ indicate covariates not directly derived from genotypic data ("non-genetic covariates"). For genotypes $g_{ij} \in \{0, 1, 2\}$ the assumptions of linearity and additivity are not restrictive. On the other hand, a typical GWAS also assumes that the covariates are linearly associated with the phenotype. This is a far more restrictive assumption if any of the covariates are continuous. $\varepsilon = (\varepsilon_i)_{i=1}^n$ is an $n \times 1$ residual vector that models the environmental effects and measurement noise.

To perform a GWAS, each variant is individually tested for association with the phenotype. For example, the $j$th variant is tested for association using the following model:

$$Y = \bar{G}_{.j}\beta_j + \tilde{X}\tilde{\gamma} + \varepsilon \qquad (5)$$

Here $\tilde{X}$ contains the known set of covariates (e.g. age and sex), in addition to adjustments for confounding that become necessary when the genotypes at SNPs $\tilde{j} \neq j$ are omitted from the model shown in Eq. (4). Confounding due to the presence of genetically related subgroups within the sample, for example subgroups of individuals with common ancestry, is referred to as population structure, and is commonly accounted for by including the top several genetic PCs in $\tilde{X}$[10,11,52].

The model in Eq. (5) can be simplified by projecting away the covariates[18,53]. Define $P = I - \tilde{X}(\tilde{X}^T \tilde{X})^{-1} \tilde{X}^T$, which is the projection onto the orthogonal complement of the linear subspace spanned by $\tilde{X}$. Multiplying Eq. (5) through by $P$ on the left yields:

$$PY = P\bar{G}_{.j}\beta_j + \varepsilon^*. \qquad (6)$$

The projected phenotype $PY$ is the residual from regression of $Y$ on $\tilde{X}$. Likewise, $P\bar{G}_{.j}$ is the residual from regression of $\bar{G}_{.j}$ on $\tilde{X}$. Importantly, if $\bar{G}_{.j}$ and $\tilde{X}$ are dependent, which is necessarily true if $\tilde{X}$ contains confounders of the genotype-phenotype relationship, then $P\bar{G}_{.j}$ will differ from $\bar{G}_{.j}$. Consequently, an analysis that residualizes only $Y$ with respect to $\tilde{X}$ will be misspecified. Instead, to remove dependence on $\tilde{X}$, both $Y$ and $\bar{G}_{.j}$ should be residualized in Eq. (5).

Though including genotypic PCs can control for population structure, it fails to correct for cryptic or family relatedness between individuals[26,27,54,55]. LMMs were introduced to GWAS to overcome these limitations[18,26-33]. LMMs account for random variation in the phenotypic mean that is correlated with the genetic relatedness of the individuals under study, and have proven effective for increasing power even when the kinship among subjects is more distant[18,32,33]. We use BOLT-LMM[18,33] to perform our analyses and we refer to it as the Baseline method.

**DeepNull model**. In this work, we consider a model in which the covariates have potentially non-linear effect on the phenotypes. The corresponding generative model for an individual $i$ can be written as

$$y_i = \bar{G}_{i.}\beta + f(X_{i.})\gamma_f + \varepsilon_i$$

where all variables are defined identically as in formula (4), $f : \mathbb{R}^q \to \mathbb{R}$ is any

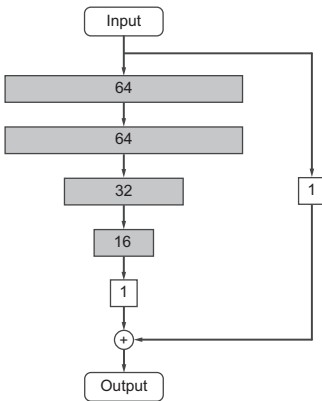

**Fig. 5 DeepNull DNN model architecture. Each rectangle represents one layer and all layers are fully connected.** Shaded layers use the ReLU activation and the non-shaded layers do not use an activation function (i.e. linear connection). The input is the set of known covariates and the output is the predicted phenotype.

(potentially non-linear) function, $\bar{G}_{i\cdot} = (\bar{g}_{ij})_{j=1}^{m}$, and $X_{i\cdot} = (x_{ik})_{k=1}^{q}$. In vector form,

$$Y = \bar{G}\beta + F(X)\gamma_f + \varepsilon$$

where $F : \mathbb{R}^{n \times q} \to \mathbb{R}^n$ is the function that applies $f$ to each row of $X$.

We convert the estimation of $u_i = f(X_{i\cdot})$ into a learning problem, where we predict $u_i$ using $y_i$ and $X_{i\cdot}$ as targets and input features, respectively. In other words, we train a model $h$ using the covariates $X_{i\cdot}$ and the phenotype $y_i$ by minimizing

$$\| y_i - h(X_{i\cdot}) \|^2. \tag{7}$$

We designed a DNN architecture for modeling the function $h$ (Fig. 5). We explored the model proposed previously to detect interpretable statistical interactions[56] but found that a simpler model with an explicit linear effect performed equally well on four UKB phenotypes tested (data not shown). The resulting model is inspired by residual networks[57] and consists of two components. One component (the shorter path from input to output in Fig. 5) is linear, to directly represent the linear effect of the covariates on the phenotype. The other component (the longer path in Fig. 5) is a multi-layer perceptron (MLP), to model a potentially non-linear effect of the covariates. The MLP component has 4 hidden layers, all of which use the Rectified Linear Unit (ReLU) activation.

In an equation form, the DeepNull model $h$ can be written as

$$h(X_{i\cdot}) = H^{(5)} + H^{(6)},$$

where

$$H^{(1)} = \phi(W_{64 \times q}^{(1)} X_{i\cdot} + B_{64 \times 1}^{(1)})$$
$$H^{(2)} = \phi(W_{64 \times 64}^{(2)} H^{(1)} + B_{64 \times 1}^{(2)})$$
$$H^{(3)} = \phi(W_{32 \times 64}^{(3)} H^{(2)} + B_{32 \times 1}^{(3)})$$
$$H^{(4)} = \phi(W_{16 \times 32}^{(4)} H^{(3)} + B_{16 \times 1}^{(4)})$$
$$H^{(5)} = W_{1 \times 16}^{(5)} H^{(4)} + B_{1 \times 1}^{(5)}$$
$$H^{(6)} = W_{1 \times q}^{(6)} X_{i\cdot} + B_{1 \times 1}^{(6)}$$

and $\phi$ is the coordinate-wise ReLU function, i.e.

$$\phi\left((x_p)_{p=1}^{P}\right) = \left(\max(0, x_p)\right)_{p=1}^{P}.$$

DeepNull learns

$$W = \{W_{64 \times q}^{(1)}, W_{64 \times 64}^{(2)}, W_{32 \times 64}^{(3)}, W_{16 \times 32}^{(4)}, W_{16 \times 32}^{(4)}, W_{1 \times 16}^{(5)}, W_{1 \times q}^{(6)}\}$$

and

$$B = \{B_{64 \times 1}^{(1)}, B_{64 \times 1}^{(2)}, B_{32 \times 1}^{(3)}, B_{16 \times 1}^{(4)}, B_{1 \times 1}^{(5)}, B_{1 \times 1}^{(6)}\}$$

by minimizing the mean squared error in (7) using the Adam optimizer[58] implemented in Keras for TensorFlow 2. Adam is run with $\beta_1 = 0.9$ and $\beta_2 = 0.99$. We also used a batch_size of 1024 and a learning_rate of $10^{-4}$. We train DeepNull for 1,000 epochs (running DeepNull with more epochs can improve the results with the cost of increasing the training time), without early stopping, batch normalization, or dropout. Kernel initializers were set to default (glorot_uniform) and bias initializers were set to default (zeros).

**Performing GWAS using DeepNull.** After training DeepNull, we use the following model to test for association between the $j$th variant and the phenotype:

$$y_i = \bar{g}_{ij}\beta_j + h(X_{i\cdot})\gamma_h + \tilde{X}_{i\cdot}\gamma + \varepsilon.$$

The vectorized form of the above association test is

$$Y = \bar{G}_{\cdot j}\beta_j + H(X)\gamma_h + \tilde{X}\gamma + \varepsilon. \tag{8}$$

Where $H : \mathbb{R}^{n \times q} \to \mathbb{R}^n$ is the function that applies $h$ to each row of $X$. Compared to the standard GWAS association model in Eq. (5), the DeepNull association model differs only by the inclusion of an extra term $H(X)\gamma_h$, where $h(X_{i\cdot})$ is the DNN's prediction of the phenotype, based on non-genetic covariates only, and $\gamma_h$ is a scalar association coefficient. As in the model shown in Eq. (5), $\tilde{X}$ includes both non-genetic covariates (e.g. age and sex) and adjustments for confounding (e.g. genetic PCs) while $X$ excludes PCs. PCs are excluded because the aim of DeepNull is to predict phenotypes without utilizing genetic data, whereas the PCs are computed from genotypes. In addition, higher order interactions of PCs may capture true biological signals that it is not desirable to remove (e.g. conditional associations) in GWAS.

To avoid overfitting, DeepNull should be trained and run on distinct sets of individuals. However, to maximize the GWAS's statistical power, all individuals in the cohort should receive DeepNull predictions. To satisfy both of these criteria, we split the cohort by individual into $k$ partitions. For each selected partition, we train a DeepNull model using data from $k - 2$ of the other partitions and use the remaining partition for validation and model selection. The model that performs best on the validation partition is then used to predict all individuals in the selected partition. The partitioning scheme ensures that each partition is used as the validation/selection partition exactly once.

**Simulation framework.** We simulate data using the model

$$Y = \bar{G}\beta + \sum_{k=1}^{q} f(X_{\cdot k})\gamma_k + \varepsilon \tag{9}$$

where $X_{\cdot k}$ is the value of the $k$-th covariate for all individuals, $\gamma_k$ is the effect size, and $f(\cdot)$ is an arbitrary function from $\mathbb{R}$ to $\mathbb{R}$, such as the identity $f(x) = x$ or exponential function $f(x) = \exp(x)$. For $j = 1, \cdots, m$, the variant effect sizes $\beta_j$ are drawn independently from a normal distribution with mean zero and variance equal to $\frac{\sigma_g^2}{m}$ where $\sigma_g^2 \in [0, 1)$ is the proportion of phenotypic variance explained by genotype (i.e., the heritability) and $m$ is the number of causal variants: $\beta_j \overset{iid}{\sim} \mathcal{N}(0, \frac{\sigma_g^2}{m})$. Similarly, the covariate effects are drawn independently from a normal distribution with mean zero and variance equal to $\frac{\sigma_x^2}{q}$ such that $\sigma_x^2$ is the proportion of phenotypic variance explained by the covariates: $\gamma_k \overset{iid}{\sim} \mathcal{N}(0, \frac{\sigma_x^2}{q})$. Lastly, $\varepsilon$ is drawn from another independent normal distribution with mean 0 and variance $1 - (\sigma_g^2 + \sigma_x^2)$: $\varepsilon \sim \mathcal{N}(0, 1 - \sigma_g^2 - \sigma_x^2)$. Under this model, $\mathbb{E}(Y) = 0$ and $\mathbb{V}(Y) = \mathbb{E}(Y^2) = 1$. In the case $f(\cdot)$ is the identity function $f(x) = x$, our simulation framework is similar to previous works[18,32].

Phenotypes were simulated based on genotypes and covariates from the UKB. Age, sex, and genotype_array were included as covariates. Causal variants were selected uniformly at random from chr22 such that 1% variants (i.e., 127 variants) were causal. Association testing was performed using BOLT-LMM[33] applied to chromosomes chr1, chr2, and chr22. BOLT-LMM is a linear mixed model that incorporates a Bayesian spike-and-slab prior for the random effects attributed to variants other than that being tested. The prior allows for a non-infinitesimal genetic architecture, in which a mixture of both small and large effect variants influence the phenotype. Specifically, the BOLT-LMM association statistic arises from Eq. (8) with the inclusion of an additional random effect $\bar{G}^{(-j)}\delta$. Here $\bar{G}^{(-j)}$ denotes genotype at all variants not on the same chromosome as variant $j$, and the components of $\delta$ follow the spike-and-slab prior[18].

In our setting, chr1 and chr2 are utilized to compute the type I error of the association test, which is the proportion of non-causal variants erroneously associated with the phenotype at a given significance threshold $\alpha$ (e.g. $\alpha = 0.05$). For null SNPs, the expected $\chi^2$ statistic is 1. Methods that effectively control type I error are compared with respect to their power for correctly rejecting the null hypothesis[59–61], and their expected $\chi^2$ statistics[18,32,33]. Power is defined as the probability of correctly detecting that a variant with a non-zero effect size is causal[59–61]. Additionally, the expected $\chi^2$ statistic of an association method is a proxy for its prediction accuracy[18,32,33].

**UKB GWAS evaluation.** All GWASs were performed in a subset of UKB individuals of European genetic ancestry, identified as in Alipanahi et al.[19]. Briefly, the medioid of the top 15 genetic PC values of all individuals with self-reported "British" ancestry was computed, then the distance from each individual in UKB to the British medioid was computed and all individuals within a distance of 40 were retained. The threshold of 40 was selected based on the 99th percentile of distances of individuals who self-identify as British or Irish.

Association testing was performed via BOLT-LMM[18,33] (Code Availability) with covariates specific to each experiment. GWAS "hits" were defined as genome-

wide significant (i.e. $P \leq 5 \times 10^{-8}$) lead variants that are independent at an $R^2$ threshold of 0.1. Hits were identified using the $--$clump command in PLINK (Code Availability). The linkage disequilibrium (LD) calculation was based on a reference panel of 10,000 randomly sampled unrelated subjects of European ancestry from the UKB. The span of each hit was defined based on the set of reference panel variants in LD with the hit at $R^2 \geq 0.1$. GWAS "loci" were defined by merging hits within 250 Kbp.

Comparison of two GWAS results $G_1$ and $G_2$ for shared and unique hits was performed by examining overlap of the hit spans; a given hit $H_1$ from $G_1$ is classified as shared if the span of any hit from $G_2$ overlaps it, otherwise it is classified as unique.

Comparison of our GWAS with the GWAS catalog (Code Availability) was performed analogously to comparing two GWAS. We used gwas_catalog_v1.0.2 $-$associations_e100_r2021$-04-05$ and converted coordinates from GRCh38 to GRCh37 using UCSC LiftOver (Code Availability) with default parameters. All catalog variants whose "DISEASE/TRAIT" column matched the phenotype of interest and were genome-wide significant were converted into loci by merging variants within 250 Kbp.

**Learning phenotype-covariates relationship via spline regression**. We can learn the non-linear relationship between the phenotype and covariates by fitting sex-specific spline regression models to predict the desired phenotype using a set of covariates. For each sex, we learn an independent spline regression model based on the other non-genetic covariates. We utilized the python scikit-learn package (Code Availability) to perform spline fitting.

**Learning phenotype-covariates relationship via XGBoost**. We can also learn the non-linear relationship between the phenotype and covariates by fitting gradient boosted decision trees. XGBoost (Code Availability) is one existing implementation of gradient boosted decision trees. We utilized XGBoost to learn the non-linear relationship. The optimal XGBoost hyperparameters were selected by performing black-box hyperparameter optimization with Google Vizier[62]. The optimization objective was to minimize root mean squared error for the totalprotein phenotype in UKB. The dataset was randomly split into train (80%) and test (20%) folds. The optimal parameters identified, and used for all 10 UKB phenotypes, were the following: max_depth = 3, eta = 0.3190, alpha = 0.6577, and lambda = 2.

**Reporting summary**. Further information on research design is available in the Nature Research Reporting Summary linked to this article.

## Data availability

This work used genotyped and phenotypes from the UK Biobank study (https://www.ukbiobank.ac.uk) and our accessed was approved under application 65275. We utilized GWAS Catalog (https://www.ebi.ac.uk/gwas/) for replication analysis. Source data are provided with this paper.

## Code availability

DeepNull software is available for download from GitHub (https://github.com/google-health/genomics-research/tree/main/nonlinear-covariate-gwas) or installation via PyPI (https://pypi.org/project/deepnull/). We used the following tools: BOLT-LMM (https://data.broadinstitute.org/alkesgroup/bolt-lmm), S-LDSC (https://data.broadinstitute.org/alkesgroup/ldscore), PLINK, scikit-learn (https://scikit-learn.org/stable/), TensorFlow (https://www.tensorflow.org), UCSC LiftOver (https://genome.ucsc.edu/cgi-bin/hgLiftOver), and XGBoost (https://xgboost.readthedocs.io/en/latest/).

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

## Acknowledgements

This research has been conducted using the UK Biobank Resource application 65275. We are grateful to Alkes L. Price for helpful comments on the manuscript. We are extremely thankful for Babak Behsaz's contribution to our in-house GWAS pipeline and Justin Cosentino for insightful comments and discussion regarding neural network interpretability.

## Author contributions

C.Y.M. and F.H. conceived the study. Z.R.M., B.A., C.Y.M., and F.H. designed the study. Z.R.M., T.C., T.Y., N.F., C.Y.M., and F.H. performed experiments. Z.R.M., T.C., T.Y., N.F., A.C., B.A., C.Y.M., and F.H. analyzed results. Z.R.M., C.Y.M., and F.H. wrote the manuscript. All authors contributed to the final version of the manuscript.

## Competing interests

All authors are employees of Google LLC. This study was funded by Google LLC.

## Additional information

**Peer review information** Nature Communications thanks Nick Shrine and the other anonymous reviewer(s) for their contribution to the peer review this work. Peer reviewer reports are available.

