## [Peer Review File · Nature Communications]

Reviewers' Comments:

Reviewer #1:

Remarks to the Author:

In this manuscript, Hormozdiari et al. proposed a machine learning method to adjust for higher-order confounding effects in genome-wide association studies (GWAS). They utilized deep neural networks (DNN), named DeepNull, to summarize various non-genetic prognostic factors to a single prediction term, and demonstrated the inclusion of this term improved GWAS statistical power and phenotype prediction.

Although the idea in this work is simple and could potentially be effective in practice, some technical details are questionable and need justifications. Furthermore, current analyses performed by the authors are only scratching the surface for their deep learning model. My comments are below.

Major:

1. The comparisons of DeepNull to baselines are incomplete to justify the use of neural networks. Currently the authors only compare to the plain/linear version, but the comparison should be performed to other machine learning methods, which may either perform better than DNN, train faster, and/or be more interpretable. I suggest learning the h function by at least a range of other machine learners SVM, random forest, xgboost, and by varying the architecture of DeepNull.

2. Even using deep neural networks, the authors should account for the interpretability of DeepNull, which currently is a crucial missing piece. In the author's context, distilling the trained model to a simple polynomial equation is particularly desirable and potentially more generalizable, e.g. in symbolic regressions. Besides, there exists many tools to open the black box for a DNN, such as DeepLift and Shap.

3. The simulation setup is incomplete. It will be useful to compare DeepNull, simpler and baseline models when there are hidden/unseen confounders with varying confounding strengths, i.e. confounding variables in the generative mechanisms but are missing in observed list of variables; and vice versa, when there are non-confounding factor but included in the input features to DeepNull.

Minor:

4. There is no evaluation information for DeepNull's prediction accuracy by non-genetic factors. How is k determined for each GWAS? What's the variation of prediction performance across different folds?

5. Related to 4, does the prediction accuracy of DeepNull using non-genetic factors relate to the gain in GWAS power and/or phenotype prediction?

6. While the number of significant hits increased by DeepNull, are the gained hits marginally significant in baseline, and/or if there are any lost hits after DeepNull?

7. More biological/functional assessment of the gained hits should make the author's argument stronger in real data applications.

8. Please elaborate how PRS_DeepNull is generated, because the description in main text seems different from the table S11 legend. The conclusion for "To determine whether the improved predictive power stems from more accurate GWAS effect size estimates or inclusion of the DeepNull DNN prediction" is also missing or confusing.

9. The average improvements in UKB GWAS are misleading because the majority of the signals are from GRP; please also provide the median.

Reviewer #2:

Remarks to the Author:

The paper proposes to use a non-linear model for covariates in GWAS and phenotype prediction.

page 2 : I do not think it is accurate to claim that GWAS have only just started to model quadratic or first-order interaction terms. I think this is prevalent and often some model exploration is done at the outset to align on a set of reasonable covariates.

The results of the Second-order Baseline model are just as good as the DeepNull results, and this is buried at the end of the paper. I think this needs to be made more prominent, as the second-order baseline is simple method for people to use. This should be done for both the association scans and the phenotype prediction parts of the paper.

Also, counting the number of associations in Supp Tables 9-10 is not completely useful. At these loci a scatter plot of the association statistics would be useful.

Looking at Supp Figs 4 it seems that a simple smooth estimate of the effect of Age on the traits for each sex might go a long way. This is simple and quick to do, and would be a valuable addition to the methods comparison here.

Reviewer #3:

Remarks to the Author:

This is a fairly straightforward methodological paper in which the authors have applied a deep neural network method to model non-linear non-genetic covariate effects and non-genetic covariate interactions in a genome-wide association study when the appropriate non-linear function or interactions are not known a priori.

They demonstrate the improved performance of their DeepNull method over the usual "Baseline" models through simulation and application to real phenotype and genotype data from UK Biobank, showing that DeepNull increases power in the presence of non-linear covariate effects and covariate interaction effects while maintaining Type I error rate and also improving phenotype prediction in polygenic risk scores.

The manuscript is clear and concise and as presented the advantages of the method are convincing. The limitations of the method are acceptable.

My concern with a paper describing new software is that the software is actually available and useable at the time of publication. This appears to be the case here as I have downloaded the software and been able to run it on my GWAS data and indeed observe some of the claimed increase in power over an a priori covariate model specified in the usual way.

It might be helpful for users of the software to have some explicit command-line examples of how to re-run Plink and BOLT-LMM incorporating the deepnull predictors into the association model.

Response to reviewers for NCOMMS-21-19339-T

Reviewer #1 (Remarks to the Author):

In this manuscript, Hormozdiari et al. proposed a machine learning method to adjust for higher-order confounding effects in genome-wide association studies (GWAS). They utilized deep neural networks (DNN), named DeepNull, to summarize various non-genetic prognostic factors to a single prediction term, and demonstrated the inclusion of this term improved GWAS statistical power and phenotype prediction.

Although the idea in this work is simple and could potentially be effective in practice, some technical details are questionable and need justifications. Furthermore, current analyses performed by the authors are only scratching the surface for their deep learning model. My comments are below.

Major:

(Reviewer #1) 1. The comparisons of DeepNull to baselines are incomplete to justify the use of neural networks. Currently the authors only compare to the plain/linear version, but the comparison should be performed to other machine learning methods, which may either perform better than DNN, train faster, and/or be more interpretable. I suggest learning the h function by at least a range of other machine learners SVM, random forest, xgboost, and by varying the architecture of DeepNull.

We thank the reviewer for this useful suggestion. It is worth mentioning that the purpose of the DeepNull model is to learn the relationship that exists among the phenotype and a set of covariates. For convenience, this relationship is often assumed to be linear, yet in reality it may be complex, including nonlinear functional forms and higher order interactions. We have shown that the power of subsequent GWAS can be improved by using DeepNull to model the relationship that exists among the phenotype and covariates. In order for this approach to be broadly applicable across the spectrum of phenotypes one might encounter in practice, the model employed by DeepNull should be sufficiently flexible and expressive to model a wide variety of functional forms. Although a less expressive model (e.g. an SVM or random forest) might suffice in particular cases, having a more expressive model (e.g. a DNN) is desirable in general. Moreover, as we have shown, even if the relationship among the phenotype and the covariates is linear, such that the additional flexibility of a DNN is not required, we do not observe a loss of power by applying DeepNull.

To further explore how other models might compare with DeepNull in practice, we examined the performance of SVM, random forest, and gradient boosted decision trees (GBDT) for predicting the 10 UKB phenotypes (Table 1 in the reviewer response). We use XGBoost as one particular piece of software that implements GBDT. We use MSE (mean

squared error) and MAE (mean absolute error) measurements to compare different methods. XGBoost tends to have the smallest MSE for all phenotypes. Mirroring DeepNull, we performed GWAS including the XGBoost phenotypic prediction (Supplementary Tables 16, 17, 20) as an additional covariate. The results were similar to using DeepNull’s prediction.

Again, it is important to emphasize four main points:

- 1) There are no existing methods that utilize ML prediction (gradient boosted decision trees such as XGBoost or otherwise) to improve power for GWAS.
- 2) DNN can handle complex inputs such as image and text while GBDT (e.g., XGBoost) typically cannot.
- 3) DeepNull does not claim to be the only method, or necessarily the most accurate method, for describing the non-linear relationship between phenotype and covariates. Our primary goal is not to generate “best” phenotypic prediction, but rather to improve upon the current state-of-the-art for performing GWAS.
- 4) We analyzed 10 real phenotypes as a proof-of-concept. Similar performance on these phenotypes does not mean that DeepNull and XGBoost will perform similarly for all phenotypes. Since DNNs are universal approximators (Hornik 1991; Leshno 1993), DeepNull should be able to approximate the phenotype-covariate relationship for any phenotype encountered in practice.

Pheno	SVM		Random Forest		XGBoost	
	MSE	MAE	MSE	MAE	MSE	MAE
ALT	189.9515	8.7663	180.7120	8.2093	180.5931	8.2057
ALP	669.8633	17.8769	659.4248	17.4900	659.1724	17.4860
Apob	0.0583	0.1896	0.0545	0.1849	0.0543	0.1846
AST	109.7121	5.2969	106.6504	5.5215	106.6455	5.5204
Calcium	0.0403	0.1630	0.0085	0.0708	0.0085	0.0707
GRP	0.0043	0.0231	0.0011	0.0122	0.0010	0.0114
LDL	0.7162	0.6738	0.7165	0.6730	0.7154	0.6724
Phosphate	0.0553	0.1862	0.0238	0.1218	0.0238	0.1218
SHBG	646.4755	17.8069	627.2297	18.1200	626.6411	18.1068
Triglycerides	1.0036	0.7474	0.9814	0.7110	0.9812	0.7107

Table 1. Comparison of SVM, Random Forest, and XGBoost on 10 UKB phenotypes. We reported the MSE and MAE among the 3 methods.

Finally, to enable users to apply different non-linear modeling techniques, we have 1) updated the open-source DeepNull software to natively support XGBoost models and DeepNull architecture changes, and 2) document a model API for further extensions.

We added the following paragraph to the discussion (p. 15):

“Lastly, DeepNull is a proof of concept. For some phenotypes, a simpler model such as the Second-order Baseline model may suffice to capture the phenotype-covariate relationship. For others, an alternative non-linear model such as boosted trees may equal or possibly outperform DeepNull’s DNN. For example, we observed that XGBoost obtained similar GWAS hits, loci, and phenotypic predictions for the 10 example UKB phenotypes (Supplementary Tables 16, 17, and 20). Although XGBoost and DNN performed similarly for these phenotypes, the added flexibility of DNNs may prove advantageous for other phenotypes or sets of covariates. For example, DNNs can handle complex inputs such as image and text that XGBoost typically cannot. Importantly, we observed in all cases that DeepNull performed as well or better than current standard practice, and the underlying DNN is sufficiently expressive to capture many of the phenotype-covariate relationships likely to be encountered in practice.”

Regarding varying the architecture of DeepNull, we initially implemented the model proposed by Tsang et al. (2018) to detect possible feature interactions. We observed that we can simplify the architecture as proposed in our paper. We compared the results of DeepNull and Tsang et al. for ApoB, AST, GRP, and LDL (4 phenotypes with the highest improvement for DeepNull). We did not observe a significant difference between the two models so we decided to use the simpler model proposed in our paper (Table 2 in the reviewer response).

	Tsang et al. ICLR 2018	DeepNull
ApoB	0.167	0.166
AST	0.200	0.199
GRP	0.842	0.850
LDL	0.210	0.217

Table 2. Comparison of DeepNull and Tsang et al. ICLR 2018 network on 4 top phenotypes. We compare the Pearson correlation of predicted and true values. We use the 5-fold cross validation prediction of each method.

We have added the following sentence to the methods (p. 18) to make this clear:

“We explored the model proposed previously to detect interpretable statistical interactions (Tsang et al 2018) but found that a simpler model with an explicit linear effect performed equally well on four UKB phenotypes tested (data not shown). The resulting model...”

(Reviewer #1) 2. Even using deep neural networks, the authors should account for the interpretability of DeepNull, which currently is a crucial missing piece. In the author's context, distilling the trained model to a simple polynomial equation is particularly desirable and potentially more generalizable, e.g. in symbolic regressions. Besides, there exists many tools to open the black box for a DNN, such as DeepLift and Shap.

DeepNull improves both GWAS power and phenotype prediction. As described above, the purpose of the DNN is to provide a flexible model for learning the potentially complex relationship that exists among the phenotype and non-genetic covariates. In the case of GWAS, this relationship is not itself of interest during subsequent analysis. Practically, we simply add a single additional covariate (the DeepNull prediction) to the current state-of-the-art GWAS. The parameter of interest is instead the coefficient describing the relationship between genotype and the phenotype. By estimating this coefficient within a linear model that incorporates DeepNull's prediction of phenotype, we obtain a more precise estimate of the genetic effect. For the aim of improving power to detect genetic signals, it is desirable for the model describing the relationship between the phenotype and covariates to be as flexible as possible. While using a more interpretable model may be informative for understanding the non-genetic basis of the phenotype, it may simultaneously reduce power for uncovering the genetic basis, which is the primary goal.

In the case of phenotype prediction, interpretability of the influences of the non-genetic factors may or may not be of interest: in cases where absolute predictive accuracy is of paramount importance, a practitioner may sacrifice interpretability for a more accurate model. However, we agree with the Reviewer that the interpretability of DeepNull is much more challenging than that of a linear model in this setting. We highlight that with the revised open-source DeepNull codebase (which extends modeling strategies to support XGBoost) it is straightforward to further extend the modeling methodology to support alternative non-linear models like symbolic regression that may enable more interpretability.

Interpretability of DNNs is an area of active research in the community, with both individual-example-level interpretability methods such as DeepLIFT and SHAP, as well as methods that attempt to develop analytical models that approximate a “black box” model like a DNN (e.g. Alaa and van der Schaar 2019; Crabbe et al 2020), but we believe a deep exploration of DeepNull interpretability for phenotype prediction is outside the scope of this work. To make this clear to the reader, we have revised the Discussion to highlight the lack of interpretability of DeepNull as a limitation of our work (p. 14-15):

“Third, DeepNull’s DNN is not easily interpretable compared to less expressive models such as the Baseline model. For improving GWAS power, this is of little consequence as the parameter of interest is the coefficient describing the relationship between genotype and phenotype, and by estimating this coefficient within a linear model that incorporates DeepNull’s prediction of phenotype, we obtain a more precise estimate of this genetic effect. For more interpretable phenotype prediction, possibly at the expense of some prediction accuracy, using a non-linear model such as spline regression or generalized additive model (Hastie and Tibshirani 1990), symbolic regression (Koza 1992), or neural additive model (Agarwal et al 2020) may be beneficial. Alternatively, the trained DeepNull model can be interrogated with a variety of methods (Lundberg and Lee 2017, Shrikumar et al 2017, Alaa and van der Schaar 2019; Crabbe et al 2020) though we note that DNN interpretability is an active and evolving area of research.”

Following the Reviewer’s suggestion, to explore how post-hoc interpretability methods could be used with DeepNull, we applied SHAP (SHapley Additive exPlanations) to the DeepNull prediction of Glaucoma referral probability (GRP) using VCDR, age, sex, and genotyping array. We observed that VCDR has the highest mean |Shap Value| of the four covariates (Figure 1A response to the reviewer). We also observed that increasing the VCDR value increases the Shap value; thus, higher VCDR values result in an increased GRP compared to average GRP which is consistent with the known relationship between VCDR and Glaucoma. Furthermore, the next most important feature is age, for which younger individuals have lower Shap values; thus, younger samples result in a decreased GRP compared to average GRP which is also consistent with the known relationship between age and Glaucoma.

It is worth mentioning that random forests and XGBoost, which is a boosted version of random forests, are also not easily interpretable (Ribeiro et al. 2016, Hara et al. 2016, Tan et al. 2020). Moreover, most methods such as SHAP that are designed to interpret deep learning models are designed for random forest and XGBoost too (check TreeExplainer in SHAP method).

To conclude, for the purpose of GWAS analysis, which is the primary focus of this paper, the interpretability of the non-genetic covariate DNN is not of essential interest. For the purpose of phenotypic prediction, the DNN’s interpretability is potentially of interest. However, given the immaturity of the field of DNN interpretation, we believe that a detailed study of DeepNull’s interpretability for phenotypic prediction is outside of scope of the present work.

Figure 1. SHAP results for the GRP phenotype. A) SHAP feature importance that illustrates the mean shapley values (SHAP value). B) SHAP summary plot. Each dot per feature indicates a sample. The X-axis is the SHAP value and the Y-axis is the importance of each feature; higher features have a larger effect on the model prediction.

(Reviewer #1) 3. The simulation setup is incomplete. It will be useful to compare DeepNull, simpler and baseline models when there are hidden/unseen confounders with varying confounding strengths, i.e. confounding variables in the generative mechanisms but are missing in observed list of variables; and vice versa, when there are non-confounding factor but included in the input features to DeepNull.

We interpret this comment as a request to explore the repercussions of omitting covariates that influence the phenotype or include extraneous covariates that have no influence on the phenotype, and thank the reviewer for the suggestion. We have performed two additional experiments to address this concern:

- 1) We simulated phenotypes, using a similar framework to that presented in Figure 1 in the paper, based on age, sex, and genotype-array. However, when we applied DeepNull and Baseline, we provided only sex and genotype-array (i.e. we omit the “age” covariate that does influence phenotype). The results are presented in Supplementary Table 3 and reproduced below for convenience. Both DeepNull and Baseline controlled the type I error while providing similar power.

σ_g^2	σ_c^2	Power		E[X ²]		E[X ²] Null		Type I Error	
		DeepNull	Baseline	DeepNull	Baseline	DeepNull	Baseline	DeepNull	Baseline
0.2	0.1	0.1962 (0.0047)	0.1962 (0.0047)	1.8581 (0.0171)	1.8581 (0.0171)	1.0035 (0.0014)	1.0036 (0.0014)	0.0499 (0.0002)	0.0499 (0.0002)
0.2	0.2	0.2146 (0.0049)	0.2147 (0.0049)	1.9298 (0.0195)	1.9300 (0.0196)	1.0038 (0.0014)	1.0038 (0.0014)	0.0499 (0.0002)	0.0499 (0.0002)
0.4	0.1	0.34378 (0.0048)	0.34363 (0.0048)	2.6795 (0.0305)	2.6794 (0.0306)	1.0061 (0.0015)	1.0060 (0.0015)	0.0502 (0.0002)	0.0501 (0.0002)
0.4	0.2	0.3571 (0.0049)	0.3565 (0.0049)	2.8112 (0.0350)	2.8117 (0.0350)	1.0063 (0.0015)	1.0064 (0.0015)	0.0501 (0.0002)	0.0501 (0.0002)
0.4	0.4	0.3778 (0.0059)	0.3789 (0.0057)	3.1678 (0.0514)	3.1685 (0.0513)	1.0062 (0.0016)	1.0061 (0.0016)	0.0499 (0.0002)	0.0499 (0.0002)
0.6	0.2	0.3995 (0.0046)	0.3984 (0.0048)	3.5943 (0.0482)	3.5941 (0.0482)	1.0059 (0.0021)	1.0056 (0.0021)	0.0499 (0.0002)	0.0499 (0.0002)

Table 3. Comparison of Baseline and DeepNull models with only linear effects of covariates to phenotype with missing non-confounding covariates. We simulated phenotypes using a similar framework as Figure 1 in the manuscript main text where we used age, sex, and genotype-array. However, when we applied DeepNull and Baseline we provided only sex and genotype-array.

- 2) We simulated phenotypes, using a similar framework to that presented in Figure 1 in the paper, based on sex and genotype-array. However, when we applied DeepNull and Baseline, we provided age, sex, and genotype-array as input (i.e. we include the extraneous covariate “age” that has no influence on the phenotype). The results are presented in Supplementary Table 4 and reproduced below for convenience. Again, both DeepNull and Baseline controlled the type I error while providing similar power.

σ_g^2	σ_c^2	Power		E[X ²]		E[X ²] Null		Type I Error	
		DeepNull	Baseline	DeepNull	Baseline	DeepNull	Baseline	DeepNull	Baseline
0.2	0.1	0.2051 (0.0045)	0.205 (0.0044)	1.8871 (0.0173)	1.8872(0.0 173)	1.0038 (0.0014)	1.0038 (0.0014)	0.0499 (0.0002)	0.0499 (0.0002)
0.2	0.2	0.2303 (0.0044)	0.2306 (0.0044)	1.9956 (0.0191)	1.9958 (0.0191)	1.0041 (0.0014)	1.0041 (0.0014)	0.0499 (0.0002)	0.0499 (0.0002)
0.4	0.1	0.3521 (0.0045)	0.3525 (0.0044)	2.7333 (0.0307)	2.7337 (0.0307)	1.0060 (0.0014)	1.0060 (0.0014)	0.0501 (0.0002)	0.0501 (0.0002)
0.4	0.2	0.3724 (0.0044)	0.3722 (0.0045)	2.9337 (0.0337)	2.9342 (0.0337)	1.0061 (0.0015)	1.0061 (0.0015)	0.0501 (0.0002)	0.0501 (0.0002)
0.4	0.4	0.3989 (0.0047)	0.3991 (0.0048)	3.4916 (0.0420)	3.4918 (0.0419)	1.0044 (0.0017)	1.0043 (0.0017)	0.0497 (0.0002)	0.0497 (0.0002)
0.6	0.2	0.3995 (0.0047)	0.3983 (0.0049)	3.7491 (0.0467)	3.75141 (0.0468)	1.0045 (0.0021)	1.0050 (0.0022)	0.04982 (0.0002)	0.0498 (0.0002)

Table 4. Comparison of Baseline and DeepNull models with only linear effects of covariates to phenotype with additional covariates with no effect on phenotypes. We simulated phenotypes using a similar framework as Figure 1 where we used sex and genotype-array. However, when we applied DeepNull and Baseline we provided age,sex, and genotype-array.

We have also added a paragraph in the result section (p. 5) to make this point clear:

“Lastly, DeepNull and Baseline perform similarly both when excluding a non-confounding covariate (i.e., hidden non-confounding covariates, Supplementary Table 3) and when including an irrelevant covariate (Supplementary Table 4).”

In case the reviewer did mean to address unobserved *confounding* variables (i.e. those with influence on both phenotype *and* genotype), we note that DeepNull does not claim to solve the problem of unobserved confounding. Even a single unobserved confounder linearly related to both genotype and phenotype could introduce an arbitrary degree of bias into estimation of the genetic effect. Moreover, if there were no observed variables

on the causal pathway between the unobserved confounder and either genotype or the phenotype, this bias would be impossible to eliminate. This is not a limitation particular to the DeepNull framework, but a known problem in causal inference (Pearl 2009). Instead, the problem addressed by DeepNull is model misspecification; specifically, misspecification of the relationship between the phenotype and covariates, particularly in the setting where that relationship is complex. By using a flexible DNN to learn the relationship between the phenotype and covariates, then adjusting for this model's expectation of the phenotype during association testing, DeepNull reduces the residual phenotypic variation, increasing power for detecting an association between genotype and the phenotype.

To illustrate the problem of unobserved confounding, suppose that the true data generating process is represented by the following graph (Figure 2 response to reviewer):

Figure 2. Causal graph. G represents the genotype, Y the phenotype, X a set of observed confounders (e.g. ancestry), and U a set of unobserved confounders.

Here G represents the genotype, Y the phenotype, X a set of observed confounders (e.g. ancestry), and U a set of unobserved confounders. As a particular example, we consider a data generating process which is an extension of that used for the quadratic bias simulations in the paper. The unobserved confounder $U \sim N(1, 1)$. Given U, genotype $G \sim N(\alpha U, 1)$. Here, as in the main text, G is simulated as a continuous random variable to allow for fine control of the correlation between G and Y. The known confounder X was drawn independently of (U, G) as $X \sim N(0, 1)$. Finally, given (U, G, X), the phenotype was generated as

$$Y \sim N(\beta_0 + \beta_U U + \beta_G G + \beta_{X_1} X + \beta_{X_2} X^2, 1)$$

We emphasize that normality is assumed for analytical convenience, but that all conclusions carry through if the random variables (U, G, X, Y) are drawn from other distributions; the important feature of the data generating process is the dependence structure among the random variables.

The best possible estimate of β_G is obtained by regressing Y on (U, G, X, X^2) . We refer to this procedure as the *oracle estimator*. But suppose U is unobserved. Lacking any information on U , the best we can do is to fit a model that correctly specifies the relationships among (G, X, Y) ; namely, regressing Y on (G, X, X^2) . We refer to this as the *confounded estimator*. It is straightforward to show that the expected bias of the confounded estimator is $\alpha\beta_U / (\alpha^2 + 1)$. Intuitively, the bias depends on the strength of association between U and G , through α , and on the strength of the association between U and Y , through β_U . To validate this formula, we performed 10^3 simulations where the true genetic effect $\beta_G = 1.0$. The parameters controlling the confounding bias (α, β_U) were set to

$$(\alpha, \beta_U) \in \{(-1.0, 4.0), (-1.0, 2.0), (1.0, 2.0)\}$$

These settings illustrate reversal of the association between G and Y , elimination of the association, and magnification of the association, respectively. The simulation results are summarized in the following box plots. In each case, the oracle estimator is unbiased for the true genetic effect of $\beta_G = 1.0$. The confounded estimator is centered on $-1.0, 0.0$, and 2.0 for the settings of reversal, elimination, and magnification settings, respectively.

Figure 3. All models are biased in the absence of information on the unobserved confounder.

These simulations demonstrate that in the absence of information on the unobserved confounder U (Figure3 in response to the reviewer), even a model that correctly specifies the relationship among all remaining variables is potentially subject to an arbitrary degree of bias. None of the considerations in this subsection are specific to DeepNull, but would affect any association model (i.e. a model that relates genotype to phenotype) from which an unobserved confounder U was omitted.

Minor:

(Reviewer #1) 4. There is no evaluation information for DeepNull's prediction accuracy by non-genetic factors. How is k determined for each GWAS? What's the variation of prediction performance across different folds?

We have now added the DeepNull prediction using non-genetic factors as Supplementary Table 8. In addition, we have added Supplementary Table 9 which contains DeepNull's prediction accuracy by fold (reproduced below as Table 5 in response to the reviewer). We note that DeepNull does not need to provide a highly accurate prediction of the phenotype in order to improve GWAS performance, only a more accurate prediction than is provided by conventional linear models.

We applied DeepNull to AST, Apob, GRP, and LDL; the result of DeepNull for different folds seems to be consistent as shown below (Table 5 in response to the reviewer):

	Fold0	Fold1	Fold2	Fold3	Fold4
GRP	0.84873541	0.85055926	0.84267703	0.86037541	0.86262677
LDL	0.21411863	0.21744333	0.22157866	0.22429785	0.22368437
ApoB	0.18219328	0.18662389	0.18988389	0.19274805	0.1926298
AST	0.18927157	0.21569019	0.20558756	0.19692872	0.19980957

Table 5. DeepNull's prediction accuracy (R) by fold. We applied DeepNull to AST, Apob, GRP, and LDL; the result of DeepNull for different folds is consistent.

We also note that the open-source DeepNull implementation provides a method to check for model training consistency across data folds which is called by default within the colab notebook for running DeepNull end-to-end.

We use k=5 to perform 5-fold cross validation. It is important to note that we use 5-fold cross validation to compute the DeepNull prediction of phenotype from non-genetic covariates only, and afterwards the DeepNull prediction is added as one additional covariate to the GWAS. We do not apply GWAS to each fold individually.

We add a sentence in the result section (p. 8) to illustrate that the DeepNull result is highly consistent among the different folds:

“For all GWAS, we first verified that the DeepNull prediction was consistent across all five data folds (Supplementary Table 9).”

(Reviewer #1) 5. Related to 4, does the prediction accuracy of DeepNull using non-genetic factors relate to the gain in GWAS power and/or phenotype prediction?

Yes to both questions. Regarding GWAS power, theoretically GWAS power should increase as the accuracy with which the phenotype can be predicted from non-genetic covariates increases. To see this, suppose the generative model is $Y = \beta_G G + h(X) + \varepsilon$, where G is genotype, X is the non-genetic covariates, $h(\bullet)$ is an unknown function, and ε is a residual with mean 0 and variance σ^2 . The power to reject the null hypothesis $H_0: \beta_G = 0$ is related to the non-centrality parameter (NCP), which is directly proportional to the sample size, and inversely proportional to the residual variance σ^2 . Intuitively, as σ^2 increases, the power to reject $H_0: \beta_G = 0$ decreases.

Now, because $h(\bullet)$ is unknown, during association testing we fit the working model $Y = \beta_G G + h^*(X) + \varepsilon$, where $h^*(X)$ is typically a simple function such as $h^*(X) = \beta_{X1}X + \beta_{X2}X^2$. We can write the working model as $Y = \beta_G G + h(X) + \varepsilon^*$, where $\varepsilon^* = h^*(X) - h(X) + \varepsilon$. The residual variance of the working model is $V(\varepsilon^*) = V\{h^*(X) - h(X)\} + \sigma^2$. Observe that the residual variance is increased by a term $V\{h^*(X) - h(X)\}$ that depends on the discrepancy between the true $h(\bullet)$ and the working $h^*(\bullet)$. If $h^*(\bullet)$ is in fact $h(\bullet)$, then the working model is correctly specified, $V\{h^*(X) - h(X)\} = 0$, and the test of $H_0: \beta_G = 0$ has optimal power. However, when $h^*(\bullet) \neq h(\bullet)$, the residual variance of the working model is *increased*, and the power to reject $H_0: \beta_G = 0$ is *decreased*. Thus, the more closely $h^*(\bullet)$ approximates $h(\bullet)$, the better the power. DeepNull uses a flexible DNN to learn $h^*(\bullet)$. By more closely approximating $h(\bullet)$ than the simple quadratics often used in practice, DeepNull is expected to reduce residual variance and thereby improve power.

We have added this discussion to the Supplementary Notes.

Regarding phenotype prediction, again supposing the generative model is $Y = \beta_G G + h(X) + \varepsilon$, by more closely modeling $h(\bullet)$ we directly improve phenotype prediction (Supplementary Table 19).

(Reviewer #1) 6. While the number of significant hits increased by DeepNull, are the gained hits marginally significant in baseline, and/or if there are any lost hits after DeepNull?

We added multiple scatter plots of $-\log p$ -values for DeepNull vs Baseline and Second-order for the 10 UKB phenotypes (Supplementary Figures 16-25). Below, we provide the scatter plot of $-\log p$ -values for DeepNull vs Baseline for the GRP phenotype. It is clear that some of the new variants that are genome-wide significant were marginally significant in Baseline (Figure 4 in response to the reviewer).

Figure 4. Comparison of DeepNull vs Baseline significance levels. The X-axis is the $-\log$ p-value of Baseline and the Y-axis is the $-\log$ p-value of DeepNull.

Plotted points in all associated figures represent “either-GWAS-significant variants”. Briefly, variants from each GWAS are subset to independent genome-wide significant “hits” based on p-value and LD (Methods). To avoid biasing results toward either GWAS, all variants that are identified as hits in either GWAS are shown. Note that multiple variants physically near each other, or in high LD, may both be plotted and consequently lead to different numbers of shared and uniquely significant variants than are reported in the hit and locus replication Supplementary Tables. In case of GRP, there are 23 either-GWAS-significant variants that were significant for Baseline that did not reach significance in DeepNull or the Second-order Baseline. Meanwhile, DeepNull detected 33 either-GWAS-significant variants that were not significant in Baseline (Figure 4 in the response to the reviewer).

Additionally, there is evidence that the additional hits detected by DeepNull are biologically relevant. FUMA gene-set enrichment analysis for GRP detected 42 significant gene sets among DeepNull’s results, predominantly related to pigmentation. In sharp contrast, no gene sets were enriched among Baseline’s results.

In the case of LDL, Baseline detected 27 either-GWAS-significant variants that were not significant in DeepNull while DeepNull detected 69 either-GWAS-significant variants that were not significant for Baseline. Not only did DeepNull detect more loci, but DeepNull had a higher replication rate of known loci from the GWAS catalog than Baseline (Supplementary Figure 14).

Finally, like any statistic, a *P*-value is a random variable (Lambert et al 1982), has a distribution, and will vary stochastically across data sets and association methods. When comparing different methods, some hits may be gained or lost due to chance. However, a method that is more powerful is still expected to have more hits on average, which is indeed the case when comparing DeepNull with Baseline (main text, Table 1).

(Reviewer #1) 7. More biological/functional assessment of the gained hits should make the author's argument stronger in real data applications.

We thank the reviewer for pointing out this interesting analysis. We performed gene set analysis via FUMA (Watanabe et al. 2017) for 4 phenotypes in UKB. We selected the 4 phenotypes (ApoB, AST, GRP, and LDL) for which DeepNull had the highest improvement in the case of phenotype prediction and detected associated loci and hits.

Comparing DeepNull and Baseline via FUMA, DeepNull's results were generally as or more enriched with biologically relevant gene sets. For example, we examined gene sets enriched among the LDL associations, filtering to those containing one or more of the following biologically relevant key terms: {cholesterol, LDL, lipid, HDL, triglyceride}. There were 72 significant gene sets among DeepNull's results, in comparison to 65 among Baseline's. All 65 significant gene sets detected by Baseline were also detected by DeepNull, whereas 7 additional gene sets were detected by DeepNull only. Among the 65 significant gene sets detected by both Baseline and DeepNull, the average $-\log_{10}(\text{p-value})$ was 14.3 with DeepNull vs. 13.9 with Baseline, and DeepNull detected as many or more of the genes in the gene set for 95.4% of gene sets (62 of 65).

We performed a similar analysis among the ApoB results, using the same key terms as for LDL except for the addition of "Alzheimer's". There were 78 significant gene sets among DeepNull's results, in comparison to 77 among Baseline's. Again all gene sets detected by Baseline were also detected by DeepNull, and among those gene sets detected by both, DeepNull detected as many or more of the component genes in 94.8% of cases (73 of 77). The average $-\log_{10}(\text{p-values})$ among gene sets enriched for both Baseline and DeepNull were similar: 11.08 vs. 11.07.

We add a paragraph in the result section (p. 8-9) to explain the FUMA results:

"For these three phenotypes, we further investigated the biological significance of the detected associations using FUMA [20](Supplementary Table 10). For GRP, 42 gene sets, predominantly related to pigmentation, were enriched among

DeepNull’s results, whereas none were enriched among the Baseline results. For LDL, DeepNull detected more gene sets overall (955 Baseline vs. 1000 DeepNull), although the gene sets detected by Baseline scored higher in terms of the average $-\log_{10}(\text{p-value})$ (8.60 Baseline vs. 8.38 DeepNull). However, when focusing on the subset related to lipid metabolism, DeepNull detected more gene sets (65 Baseline vs. 72 DeepNull) and did so at a higher level of significance (average $-\log_{10}(\text{p-value})$: 13.88 Baseline vs. 14.34 DeepNull). For ApoB, DeepNull detected fewer gene sets overall (983 Baseline vs. 946 DeepNull), but at a higher level of significance (average $-\log_{10}(\text{p-value})$: 7.65 Baseline vs. 7.81 DeepNull). The gene sets detected by DeepNull are related to lipid metabolism and neurological conditions, including Alzheimer’s disease.”

(Reviewer #1) 8. Please elaborate how PRS_DeepNull is generated, because the description in main text seems different from the table S11 legend. The conclusion for "To determine whether the improved predictive power stems from more accurate GWAS effect size estimates or inclusion of the DeepNull DNN prediction" is also missing or confusing.

We thank the reviewer for pointing out this confusion. We updated a paragraph in the result section (p. 10) to make the PRS generation more clear:

“The “Baseline” model includes a PRS computed using the PLINK method using $-\text{score}$ ($\text{PRS}_{\text{baseline}}$) and linear effect of covariates to phenotype ($\text{PRS}_{\text{baseline}} + \text{Linear covariates}$). The “DeepNull-Baseline” model includes a PRS computed in the same way except using association results from DeepNull ($\text{PRS}_{\text{DeepNull}} + \text{Linear covariates}$), and “DeepNull” is a model that includes both the DeepNull-based PRS and the DeepNull prediction (non-linear effect of covariates to phenotype).”

and also update the result section (p. 10) of the analysis as well:

“This model produces slightly higher R^2 compared to Baseline for seven of the ten phenotypes, though the difference is not statistically significant for any phenotype (Supplementary Table 18), indicating that most of the improved predictive power arises due to better modeling the effects of non-genetic factors.”

(Reviewer #1) 9. The average improvements in UKB GWAS are misleading because the majority of the signals are from GRP; please also provide the median.

We computed the median for percentage of phenotype improvement over all 10 real phenotypes in UKB and we observed a median of 16.08 compared to the average of 23.72. We add the median for percentage of phenotype improvement to the manuscript as well.

“Overall, DeepNull improves phenotype prediction (average improvement=23.72%, median improvement=16.08%) across the ten phenotypes analyzed (average n=370K; Supplementary Table 18).”

In addition, in the original paper we provide two ways of computing the average to avoid this bias toward GRP results: 1) the average improvement and 2) the improvement computed on the average hits and loci. We added the median improvement for hits and loci as well.

“In addition, the median number of hits and loci increased by 3.5% and 3.7%, respectively. ”

It is worth noting that 3.5% and 3.7% are not much different from what was previously reported: ***“The average number of hits increased by 3.29%, from 824.4 for Baseline to 851.6 for DeepNull, and the average number of loci increased by 3.93%, from 214.1 to 222.5.”***

Reviewer #2 (Remarks to the Author):

The paper proposes to use a non-linear model for covariates in GWAS and phenotype prediction.

(Reviewer #2) page 2 : I do not think it is accurate to claim that GWAS have only just started to model quadratic or first-order interaction terms. I think this is prevalent and often some model exploration is done at the outset to align on a set of reasonable covariates.

Thank you for the observation. We have revised this sentence to say:

“The simplest form of covariate adjustment is to include a linear term for the covariate in the association model. If the phenotypic mean changes non-linearly with the covariate, the residual variation may be further reduced by including higher order adjustments, such as quadratic or interaction terms, as in the following recent examples [12–14].”

(Reviewer #2) The results of the Second-order Baseline model are just as good as the DeepNull results, and this is buried at the end of the paper. I think this needs to be made more prominent, as the second-order baseline is simple method for people to use. This should be done for both the association scans and the phenotype prediction parts of the paper.

It is correct that for most phenotypes the second-order baseline performs similarly with DeepNull in terms of GWAS power and phenotype prediction. However, this is not true in the case of Glaucoma referral probability (GRP). We observed that DeepNull’s phenotypic prediction has an R^2 of 0.7259 (0.7124, 0.7366) compared to 0.6492 (0.6371, 0.6595) for second-order baseline. For genome-wide significant hits, DeepNull detects 38 hits and

loci while second-order baseline detects 35 hits. Moreover, DeepNull's results were enriched for 42 significant gene sets, whereas no gene sets were enriched among Baseline's (please see our response to Reviewer #1, comment 7). This paper is a proof-of-concept that non-linear relationships between the phenotype and covariates exist, that in some cases (e.g. GRP) this relationship is more complicated than second-order polynomials, and that DeepNull generally improves power in the case of non-linear relationships. It is important to note as well that the second-order baseline model, while tractable in the case of 3 covariates, quickly becomes intractable with more covariates, due to multicollinearity issues, and in either case is more complex than the association models employed by many GWAS.

We added a paragraph in the introduction (p. 3) to make this point clear:

“Although simpler models (e.g., second-order interaction between covariates) may suffice in particular cases, the DNN architecture is sufficiently expressive to capture the broad range of phenotype-covariate relationships that researchers might encounter in practice.”

We also added the following paragraph to the discussion (p. 14):

“Lastly, DeepNull is a proof of concept. For some phenotypes, a simpler model such as the Second-order Baseline model may suffice to capture the phenotype-covariate relationship. For others, an alternative non-linear model such as boosted trees may equal or possibly outperform DeepNull's DNN. For example, we observed that XGBoost obtained similar GWAS hits, loci, and phenotypic predictions for the 10 example UKB phenotypes (Supplementary Tables 16, 17, and 20). Although XGBoost and DNN performed similarly for these phenotypes, the added flexibility of DNNs may prove advantageous for other phenotypes or sets of covariates. For example, DNNs can handle complex inputs such as image and text that XGBoost typically cannot. Importantly, we observed in all cases that DeepNull performed as well or better than current standard practice, and the underlying DNN is sufficiently expressive to capture many of the phenotype-covariate relationships likely to be encountered in practice.”

(Reviewer #2) Also, counting the number of associations in Supp Tables 9-10 is not completely useful. At these loci a scatter plot of the association statistics would be useful.

We thank the reviewer for the helpful suggestion. We now include the scatter plots of $-\log(p\text{-value})$ for Baseline, second-order, and DeepNull of the 10 UKB phenotypes for all variants. These plots indicate that DeepNull p-values are generally more significant than Baseline and second-order. We add the following sentence to the result section (p. 9):

“In addition, DeepNull tends to have a higher level of significance for variants compared to Baseline (Supplementary Figures 16-25).”

For example, in the case of LDL, we plot the scatter plot of association statistics between DeepNull and Baseline (Figure 5 in the response to reviewer; Supplementary Figure 22). We observed that in the case of LDL, DeepNull detected 69 significant regions that are not significant in Baseline (green dots in Figure 5) vs Baseline detected 27 significant regions that are not significant in DeepNull (orange dots in Figure 5). As described above (Reviewer #1 comment #6), plotted points in all associated figures represent “either-GWAS-significant variants”. Briefly, variants from each GWAS are subset to independent genome-wide significant “hits” based on p-value and LD (Methods). To avoid biasing results toward either GWAS, all variants that are identified as hits in either GWAS are shown. Note that multiple variants physically near each other, or in high LD, may both be plotted and consequently lead to different numbers of shared and uniquely significant variants than are reported in the hit and locus replication Supplementary Tables.

Figure 5. Significant level comparison of DeepNull vs Baseline for LDL. The orange dots indicate regions that are significant for Baseline but not significant for DeepNull and green dots indicate regions that are significant for DeepNull but not significant for Baseline.

Furthermore, we plot the scatter plot of association statistics between DeepNull and Second-order for LDL phenotype (Figure 6 in the response to the reviewer or Supplementary Figure 22). While the p-values fall much closer to the $y=x$ axis, indicating that the Second-order baseline is closer to capturing the true covariate-phenotype relationship than the linear Baseline, DeepNull detected 15 significant variants that are not significant in Second-order (green dots in Figure 6) whereas Second-order detected 10 significant variants that are not significant in DeepNull (orange dots in Figure 6).

Figure 6. Significant level comparison of DeepNull vs Second-order for LDL. The orange dots indicate regions that are significant for Second-order but not significant for DeepNull and green dots indicate regions that are significant for DeepNull but not significant for Second-order.

We do agree that visualizing the scatter plot of the association statistics can be helpful to compare different methods; however, counting the number of significant loci and hits is

widely used to compare the power of statistical methods (Darnell et al. 2012, Wen 2016, GTEx consortium 2017, Kichaev et al. 2019, GTEx Consortium 2020) so we have elected to retain the hit and locus count supplementary tables as well.

(Reviewer #2) Looking at Supp Figs 4 it seems that a simple smooth estimate of the effect of Age on the traits for each sex might go a long way. This is simple and quick to do, and would be a valuable addition to the methods comparison here.

We thank the reviewer for the suggestion. We performed Spline (smooth fitting) estimates of age for each sex. These results are added to Supplementary Tables 16, 17 and 19. We note that in the case of GRP, in addition to age and sex interaction, modeling a substantially non-linear effect of VCDR improves phenotype prediction and increases the number of discovered hits and loci. We observed that in all cases DeepNull outperforms sex-specific spline fitting of all other covariates, both in the case of phenotype prediction and number of significant hits and loci.

We add a new paragraph section in the result section (p. 9):

“Lastly, we compared the number of hits and loci of DeepNull with an extended Baseline model that performs sex-specific spline fitting (Methods) and observed that DeepNull outperforms this Baseline extension as well (compare Supplementary Tables 14 and 16 for hits and Supplementary Tables 15 and 17 for loci).”

We also added a new paragraph section in the result section (p. 10-11):

“Lastly, we compared phenotype prediction of DeepNull to an extended Baseline model that perform sex-specific spline fitting (Methods) and observed that DeepNull outperforms this Baseline extension as well (compare Supplementary Tables 18 and 19).”

In addition, we added a new subsection in the method section (p. 22):

“Learning phenotype-covariates relationship via spline regression.

We can learn the non-linear relationship between phenotype and covariates by fitting sex-specific spline regression models to predict the desired phenotype using a set of covariates. For each sex, we learn an independent spline regression model based on the other non-genetic covariates. We utilized the python Sklearn package (URLs) to perform the spline fitting.”

Reviewer #3 (Remarks to the Author):

This is a fairly straightforward methodological paper in which the authors have applied a deep neural network method to model non-linear non-genetic covariate effects and non-genetic covariate interactions in a genome-wide association study when the appropriate non-linear function or interactions are not known a priori.

They demonstrate the improved performance of their DeepNull method over the usual "Baseline" models through simulation and application to real phenotype and genotype data from UK Biobank, showing that DeepNull increases power in the presence of non-linear covariate effects and covariate interaction effects while maintaining Type I error rate and also improving phenotype prediction in polygenic risk scores.

The manuscript is clear and concise and as presented the advantages of the method are convincing. The limitations of the method are acceptable.

(Reviewer #3) My concern with a paper describing new software is that the software is actually available and useable at the time of publication. This appears to be the case here as I have downloaded the software and been able to run it on my GWAS data and indeed observe some of the claimed increase in power over an a priori covariate model specified in the usual way.

We thank the reviewer for their positive feedback.

(Reviewer #3) It might be helpful for users of the software to have some explicit command-line examples of how to re-run Plink and BOLT-LMM incorporating the deepnull predictors into the association model.

We now include an explicit command-line example contrasting a BOLT-LMM GWAS run with and without incorporating the DeepNull predictor as part of the GitHub README landing page:

<https://github.com/Google-Health/genomics-research/tree/main/nonlinear-covariate-gwas#incorporating-deepnull-into-a-gwas-analysis>.

References

Kurt Hornik. *Approximation capabilities of multilayer feedforward networks*. *Neural Networks*. 1991. doi: 10.1016/0893-6080(91)90009

Moshe Leshno, Vladimir Ya. Lin, Allan Pinkus, and Shimon Schocken. *Multilayer feed-forward networks with a nonpolynomial activation function can approximate any function*. *Neural Networks*, 1993. doi: 10.1016/s0893-6080(05)80131-5

Judea Pearl. *Causality: Models, Reasoning, and Inference*. 2009 (2nd edition). Cambridge University Press. doi: <https://doi.org/10.1017/CBO9780511803161>.

Marco Tulio Ribeiro, Sameer Singh, Carlos Guestrin 2016. "Why Should I Trust You?": *Explaining the Predictions of Any Classifier*. arXiv:1602.04938

Lambert D and Hall WJ. *Asymptotic Lognormality of P-Values*. *Ann Statist*. 1982; 10(1) 44 - 64. <https://doi.org/10.1214/aos/1176345689>.

GTEx Consortium 2017. *Genetic effects on gene expression across human tissues*. *Nature* 2017.

GTEx Consortium 2020. *The GTEx Consortium atlas of genetic regulatory effects across human tissues*. *Science* 2020.

Gleb Kichaev, Gaurav Bhatia, Po-Ru Loh, Steven Gazal, Kathryn Burch, Malika K. Freund, Armin Schoech, Bogdan Pasaniuc, Alkes L. Price. *Leveraging Polygenic Functional Enrichment to Improve GWAS Power*. *Am. J. Hum. Genet* 2017.

Xiaoquan Wen. *Molecular QTL discovery incorporating genomic annotations using Bayesian false discovery rate control*. *Annals of Applied Statistics* 2016.

Michael Tsang, Dehua Cheng, Yan Liu. *Detecting statistical interactions from neural network weights*. *International Conference on Learning Representations (ICLR)* 2018.

Sarah Tan, Matvey Soloviev, Giles Hooker, Martin T. Wells. *Tree Space Prototypes: Another Look at Making Tree Ensembles Interpretable*. *FODS* 2020.

Satoshi Hara, Kohei Hayashi. *Making Tree Ensembles Interpretable*. arXiv 2016.

Reviewers' Comments:

Reviewer #1:

Remarks to the Author:

I much appreciate the authors' efforts in addressing my comments. The quality of the MS has substantially improved.

I have some remaining questions/comments based on the updated MS:

1. The authors showed, very clearly in math, that a better prediction accuracy will lead to better GWAS power (response to Reviewer #1.5); yet, they don't show a head-to-head comparison of prediction accuracy between XGBoost and DeepNull. I found this counterintuitive and would suggest Table 1 is important information for reader and users, should be explicitly stated and referenced in the main results (instead of in discussion).

2. GRP comes as a strong example to support DeepNull in many places that the authors referred to (e.g., response to Reviewer #2.1). I have some concerns over the robustness of this example:

- XGBoost (MAE=0.114) and Random Forest (MAE=0.122, table 1 in response to reviewers) can predict GRP better than DeepNull (MAE=0.013, S Table 8 in revised MS).

- However, the hits after adjusting for XGBoost and DeepNull appeared to be very different (26 hits shared, but 11 XGBoost-only, 12 DeepNull-only; S Table 16).

Thus, it will be useful to help readers decide how to reconcile such discrepancies in hits between XGBoost and DeepNull (perhaps Random Forest as well given its high performance), where both ML models were highly accurate.

Reviewer #2:

Remarks to the Author:

The authors have answered all my questions. Congratulations on a nice paper.

Reviewer #3:

Remarks to the Author:

This is a fairly straightforward methodological paper in which the authors have applied a deep neural network method to model non-linear non-genetic covariate effects and non-genetic covariate interactions in a genome-wide association study when the appropriate non-linear function or interactions are not known a priori.

They demonstrate the improved performance of their DeepNull method over the usual "Baseline" models through simulation and application to real phenotype and genotype data from UK Biobank, showing that DeepNull increases power in the presence of non-linear covariate effects and covariate interaction effects while maintaining Type I error rate and also improving phenotype prediction in polygenic risk scores.

The manuscript is clear and concise and as presented the advantages of the method are convincing. The limitations of the method are acceptable.

My concern with a paper describing new software is that the software is actually available and useable at the time of publication. This appears to be the case here as I have downloaded the software and been able to run it on my GWAS data and indeed observe some of the claimed increase in power over an a priori covariate model specified in the usual way.

It is good to see that, as requested, they now include an explicit command-line example contrasting a BOLT-LMM GWAS run

with and without incorporating the DeepNull predictor as part of the GitHub README landing page

Response to reviewers for NCOMMS-21-19339A

Reviewer #1 (Remarks to the Author):

I much appreciate the authors' efforts in addressing my comments. The quality of the MS has substantially improved.

I have some remaining questions/comments based on the updated MS:

1. The authors showed, very clearly in math, that a better prediction accuracy will lead to better GWAS power (response to Reviewer #1.5); yet, they don't show a head-to-head comparison of prediction accuracy between XGBoost and DeepNull. I found this counterintuitive and would suggest Table 1 is important information for reader and users, should be explicitly stated and referenced in the main results (instead of in discussion).

We have merged Table 1, which was shown in response to the reviewer NCOMMS-21-19339-T, and Supplementary Table 8 to provide a clear head-to-head comparison of DeepNull and XGboost. The new merged Table is Supplementary Table 20. In addition, we have updated the results to note that DeepNull and XGBoost detected similar numbers of GWAS hits and loci, and performed comparably for phenotypic prediction. It is worth mentioning that in the previous revision, we had provided head-to-head comparisons of DeepNull and XGboost with respect to GWAS power (#loci and #hits) in Table 16 and phenotypic prediction in Supplementary Table 21 vs 18.

We would like to emphasize that deep neural networks (DeepNull) and XGBoost performed similarly on the 10 example phenotypes from the UK Biobank, and that for many of the analyses reported in our paper, either model would have sufficed. It is well known in machine learning that no single class of models uniformly dominates. For some phenotypes, deep neural networks may achieve the best predictive performance, while for others, XGBoost may be preferred. We have recommended deep neural networks as a reasonable default because these models are familiar to many practitioners, are easy to tune, and are in principle more flexible than boosted trees. The question of which model to adopt for a particular phenotype will vary case by case, and is of secondary interest to our exposition. The primary goal is to demonstrate that using a flexible non-linear approach to model the phenotype-covariate relationship, whether deep neural networks, XGBoost, or something else, improves upon the current standard practice, which is to use simple linear or quadratic adjustments.

2. GRP comes as a strong example to support DeepNull in many places that the authors referred to (e.g., response to Reviewer #2.1). I have some concerns over the robustness of this example:

- XGBoost (MAE=0.114) and Random Forest (MAE=0.122, table 1 in response to reviewers) can predict GRP better than DeepNull (MAE=0.013, S Table 8 in revised MS).

- However, the hits after adjusting for XGBoost and DeepNull appeared to be very different (26 hits shared, but 11 XGBoost-only, 12 DeepNull-only; S Table 16).

Thus, it will be useful to help readers decide how to reconcile such discrepancies in hits between XGBoost and DeepNull (perhaps Random Forest as well given its high performance), where both ML models were highly accurate.

While we note that the MAEs for XGBoost and RF were 0.011 and 0.012, we appreciate the point that for GRP, alternatives to deep neural networks may have had numerically (although here not significantly) superior predictive accuracy. The GWAS performances of DeepNull and XGBoost were in fact very similar, with the hits reaching significance in DeepNull only being very close to significance with XGBoost and conversely. To make this observation clear, we have added a new Figure below comparing the -log p-values of DeepNull and XGBoost. As acknowledged in our response to the previous comment, use of either DeepNull or XGBoost constitutes a significant improvement over the baseline, while the differences between DeepNull and XGBoost are comparatively minor. Again we find the comparison of the GWAS hits/loci between DeepNull and XGBoost of secondary importance to the comparison of hits/loci between DeepNull, the newly proposed methodology, and baseline (or second-order baseline), the existing standard.

Figure 1. Significance level comparison of DeepNull vs XGBoost for Glaucoma referral probability (GRP). The X-axis is the $-\log$ p-value of XGBoost. The Y-axis is $-\log$ p-value of the DeepNull. Both DeepNull and XGBoost p-values are computed using a two-sided test. The vertical and horizontal red line indicates the genome-wide significance level. The diagonal red line indicates the $y=x$. The orange dots indicate variants that are significant for XGBoost but not significant for DeepNull and green dots indicate variants that are significant for DeepNull but not significant for XGBoost.

Reviewer #2 (Remarks to the Author):

The authors have answered all my questions. Congratulations on a nice paper.

We thank the reviewers for their positive feedback.

Reviewer #3 (Remarks to the Author):

This is a fairly straightforward methodological paper in which the authors have applied a deep neural network method to model non-linear non-genetic covariate effects and non-genetic covariate interactions in a genome-wide association study when the appropriate non-linear

function or interactions are not known a priori.

They demonstrate the improved performance of their DeepNull method over the usual "Baseline" models through simulation and application to real phenotype and genotype data from UK Biobank, showing that DeepNull increases power in the presence of non-linear covariate effects and covariate interaction effects while maintaining Type I error rate and also improving phenotype prediction in polygenic risk scores.

The manuscript is clear and concise and as presented the advantages of the method are convincing. The limitations of the method are acceptable.

My concern with a paper describing new software is that the software is actually available and useable at the time of publication. This appears to be the case here as I have downloaded the software and been able to run it on my GWAS data and indeed observe some of the claimed increase in power over an a priori covariate model specified in the usual way. It is good to see that, as requested, they now include an explicit command-line example contrasting a BOLT-LMM GWAS run with and without incorporating the DeepNull predictor as part of the GitHub README landing page

We thank the reviewers for their positive feedback.